

# The Impact of the Stratospheric Quasi-Biennial Oscillation on Arctic Polar Stratospheric Cloud Occurrence

Douwang Li[1], Zhe Wang[1], Siyi Zhao[1], Jiankai Zhang[1]*, Wuhu Feng[2,3], Martyn P. Chipperfield[3,4]

[1]Key Laboratory for Semi-Arid Climate Change of the Ministry of Education, School of Atmospheric Sciences, Lanzhou University, Lanzhou, 730000, China.

[2]National Centre for Atmospheric Science (NCAS), University of Leeds, Leeds, LS2 9PH, UK

[3]Institute for Climate and Atmospheric Science, School of Earth and Environment, University of Leeds, Leeds, LS2 9JT, UK

[4]National Centre for Earth Observation (NCEO), University of Leeds, Leeds, LS2 9JT, UK

*Correspondence to*: Jiankai Zhang (jkzhang@lzu.edu.cn)

**Abstract.** Polar stratospheric clouds (PSCs) play a critical role in stratospheric ozone depletion. Previous studies have shown that the quasi-biennial oscillation (QBO) influences the Arctic stratospheric polar vortex and ozone, yet no studies have deeply analyzed the impact of the QBO on Arctic PSC occurrence. This study analyzes this impact using CALIPSO observations from 2006 to 2021 and SLIMCAT simulations from 1979 to 2022. The results show that the winter PSC coverage area is significantly larger during the westerly QBO (WQBO) phase than during the easterly QBO (EQBO) phase, with a zonal asymmetry in PSC occurrence frequency anomalies. The QBO influences the temperature, water vapour ($H_2O$), and nitric acid ($HNO_3$) in the Arctic stratosphere, which are key factors affecting PSC formation. During the WQBO phase, Arctic stratospheric temperatures show negative anomalies, with the centre of this anomaly biased towards North America. In addition, $H_2O$ shows positive anomalies in the Arctic lower stratosphere, mainly due to the stronger polar vortex preventing the transport of high-moisture air at high latitudes to mid-latitudes, causing $H_2O$ to accumulate inside the polar vortex. $HNO_3$ shows negative anomalies, primarily caused by denitrification through nitric acid trihydrate (NAT) sedimentation. Sensitivity analyses further indicate that the QBO-induced temperature anomalies are the main factor influencing PSC area, while the direct effect of $H_2O$ anomalies on PSCs is relatively small. The reduction of $HNO_3$ mainly affects PSCs in late winter and early spring. This work implies that future changes in the QBO may influence ozone through its impact on PSCs.

## 1 Introduction

Polar stratospheric clouds (PSCs) form at low temperatures in the lower and middle polar stratosphere during winter and early spring, and their particle surfaces provide sites for heterogeneous chemical reactions that can convert chlorine reservoir species (HCl, $ClONO_2$, etc.) into active forms (Cl, ClO, etc.). When spring arrives, these reactive chlorine atoms participate in the $Cl_x$ catalytic cycles that destroy stratospheric ozone (Solomon et al., 1986, 2015). Previous studies have shown that there is a linear relationship between ozone loss and PSC volume illuminated by sunlight. (Rex et al., 2004; Chipperfield et al., 2005). Therefore, the PSCs play an important role in the Antarctic ozone hole and Arctic stratospheric ozone depletion. Over the past four decades since the emergence of ozone depletion, much research has been conducted on PSCs (e.g. McCormick et al., 1982; Schreiner et al., 2002; Lowe and MacKenzie, 2008; Pitts et al., 2018; Voigt et al., 2018; Tritscher et al., 2021). According



to their composition and physical phase state, PSCs are classified into three basic types (Browell et al., 1990; Toon et al., 1990; Carslaw et al., 1994; Hanson and Mauersberger, 1988; Tritscher et al., 2021): nitric acid trihydrate (NAT), supercooled ternary solution (STS; Tabazadeh et al., 1994), and ice. The formation processes of these three PSC types differ significantly: the global Junge layer, located at around 20 km, is an aerosol layer composed of sulfuric acid ($H_2SO_4$) and water (Junge et al., 1961). As the temperature decreases in polar winter, these stratospheric sulfuric acid aerosols (SSA) grow by absorbing nitric acid ($HNO_3$) and water vapour ($H_2O$) until the temperature reaches about 192 K to form STS. The formation of ice can occur via two pathways: one is homogeneous nucleation at a supercooling of ~3 K below the ice frost point $T_{ice}$ (~184 K) (Carslaw et al., 1998; Koop et al., 1998), while the other is heterogeneous nucleation below $T_{ice}$ (Engel et al., 2013; Voigt et al., 2018). NAT can form on the pre-existing ice particles (Carslaw et al., 1998; Wirth et al., 1999) or on the SSA particles containing meteoritic dust (Lambert et al., 2016) or wildfire smoke (Ansmann et al., 2022). The existence temperature of NAT ($T_{NAT}$) is not too low, typically around 195 K. However, studies show that heterogeneous chlorine activation occurs predominantly on STS (Wegner et al., 2012), which does not mean that the NAT and ice are not important for ozone depletion. In contrast, NAT and ice can redistribute the $HNO_3$ and $H_2O$ in the polar stratosphere through sedimentation, which will affect further PSC formation as well as ozone depletion (Hunt, 1966; Nicolet, 1970; Salawitch et al., 1993).

The formation of PSCs is primarily influenced by temperature and concentrations of $H_2O$, $HNO_3$ and $H_2SO_4$ in the gas phase (Leroux and Noel, 2024). It is noteworthy that the temperature and chemical species in the polar stratosphere are not only influenced by polar physical and dynamical processes but also by the tropical atmospheric (Strahan et al., 2009; Bittner et al., 2016; Matsumura et al., 2021). The quasi-biennial oscillation (QBO) is the dominant mode of variability in the tropical stratosphere (Hansen et al., 2013) and one of the major external drivers of the wintertime polar vortex in the Northern Hemisphere (NH) stratosphere (Garfinkel et al., 2012). The QBO is characterized by an alternating pattern of easterly and westerly winds propagating downward through the tropical stratosphere, with a period of approximately 28 months. It is primarily driven by upward-propagating gravity waves, inertia-gravity waves, Kelvin waves, and Rossby-gravity waves (Holton and Lindzen, 1972; Plumb, 1977; Dunkerton, 1997; Baldwin et al., 2001).

Many studies based on observations or reanalysis (Holton and Tan, 1980; Lu et al., 2014; Yamazaki et al., 2020) and numerical models (Garfinkel et al., 2012; Hansen et al., 2013; Elsbury et al., 2021) indicate that the NH polar vortex is weaker during the easterly QBO (EQBO) phase compared to the westerly QBO (WQBO) phase, which is known as the "Holton-Tan (HT) effect" (Holton and Tan, 1980). Various explanations have been proposed to account for this phenomenon, which can be categorized into two mechanisms. The first mechanism, proposed by Holton and Tan, (1980, 1982), suggests that the QBO affects the NH polar vortex by adjusting the position of the critical line (i.e., zero wind line). During the EQBO phase, the critical line is shifted to the NH subtropics, which reflects more planetary waves toward the Arctic, weakening the polar vortex. In contrast, during the WQBO phase, the critical line located in the Southern Hemisphere (SH) allows more planetary waves to propagate toward the equator, resulting in a stronger Arctic stratospheric polar vortex (Lu et al., 2014). However, another mechanism suggests that the critical line mechanism is not important in the HT effect, arguing that meridional circulation induced by QBO can affect the propagation of planetary waves by influencing the wave refraction index (Garfinkel et al., 2012;





White et al., 2016). Overall, the QBO influences the strength and temperature of the NH polar vortex through the dynamical interactions between wave and mean flows. Furthermore, studies have shown that the frequency of sudden stratospheric warmings (SSW) in the Arctic is higher during the EQBO phase than during the WQBO phase (Holton and Tan, 1980; Salminen et al., 2020), which have a significant impact on the polar vortex and temperature. Since PSCs are temperature-sensitive, the QBO may have a potential influence on PSC occurrence by influencing the polar vortex.

On the other hand, the QBO also influences stratospheric chemical species through dynamical and chemical processes. Previous studies have focused on the impact of the QBO on stratospheric chemical species such as ozone, $H_2O$, and methane ($CH_4$) (Hansen et al., 2013; Tweedy et al., 2017; Zhang et al., 2021; Wang et al., 2022). Among these, the transport of dynamical processes is the primary pathway by which the QBO affects stratospheric chemical species. For instance, some studies have linked anomalies in tropical stratospheric chemical species to vertical motion anomalies induced by the QBO. During the EQBO phase, anomalous upward motion in the lower tropical stratosphere induced by the QBO leads to the transport of ozone-poor air from the troposphere into the lower stratosphere, resulting in a negative ozone anomaly in the lower tropical stratosphere (Gray and Pyle, 1989; Butchart et al., 2003; Zhang et al., 2021). In contrast, during the WQBO phase, the weakening of the upward motion results in a positive ozone anomaly in the lower tropical stratosphere. Furthermore, the QBO's influence on the tropical secondary circulation also affects the transport of other gases into the stratosphere, such as $CH_4$ (Xia et al., 2019), HCl (Yuejuan et al., 2005), and $N_2O$ (Park et al., 2017). Notably, $H_2O$ entering the stratosphere is influenced by the temperature of the tropical tropopause (cold point temperature), which can lead to the condensation and dehydration of $H_2O$ (Holton and Gettelman, 2001; Tian et al., 2019; Keeble et al., 2021). Anomalous upward motion near the tropopause during the EQBO phase leads to a decrease in the cold point temperature, thereby reducing the amount of $H_2O$ entering the stratosphere. Consequently, a negative $H_2O$ anomaly is observed in the lower tropical stratosphere during the EQBO phase (Hansen et al., 2013; Diallo et al., 2022). In addition to modulating the ascent of the Brewer-Dobson (BD) circulation in the tropics and affecting the distribution of chemical species in the tropical stratosphere, the QBO may also influence the downward branch of the BD circulation to further affect the chemical composition in high-latitude regions of the stratosphere. Hansen et al. (2013) found that, compared to the EQBO phase, the downward branch of the BD circulation in the polar regions above the tropopause is weakened during the WQBO phase, leading to less $H_2O$ being transported downward to the tropopause, which results in the accumulation and a positive anomaly of $H_2O$ in the Arctic lower and middle stratosphere.

Compared to dynamical processes, the QBO has a relatively small impact on stratospheric species through chemical processes, primarily affecting reaction rates by modulating temperature. Previous studies have shown that temperature anomalies in the lower tropical stratosphere induced by the QBO contribute to ozone anomalies to some extent (Ling and London, 1986; Zhang et al., 2021). Additionally, Zhang et al. (2021) found that during the EQBO phase, a reduction in PSCs associated with QBO-induced warming weakens heterogeneous chemical processing, leading to a decrease in active chlorine and an increase in ozone in the Arctic lower stratosphere. It is worth noting that their study primarily focused on the dynamical and chemical effects of the QBO on stratospheric ozone, without a detailed exploration of the QBO's impact on PSCs. Given that the QBO not only influences the temperature in the Arctic lower stratosphere but also affects chemical species, these key



factors can impact the formation of PSCs. However, no study has deeply investigated the processes and mechanisms of the QBO effects on PSCs. This study will use Cloud-Aerosol Lidar and Infrared Pathfinder Satellite Observations (CALIPSO) satellite PSC observations (Pitts et al., 2018) in conjunction with the SLICMAT 3-D chemical transport model to investigate the potential impact of the QBO on Arctic PSC occurrence.

## 2 Data and methods

### 2.1 Data

#### 2.1.1 CALIPSO PSC observations

As a part of the A-Train, CALIPSO provides global profiling of aerosols and clouds in the troposphere and lower stratosphere. Following its launch in 2006, and operating continuously until 2023, CALIPSO collected data nearly every day along 14−15 orbits between 82°S and 82°N (Winker et al., 2009). The Cloud-Aerosol Lidar with Orthogonal Polarization (CALIOP) aboard CALIPSO is a dual-wavelength polarization-sensitive lidar that simultaneously produces linearly polarized pulses at 532 nm and 1,064 nm and receives both parallel and perpendicular components of the 532 nm return signal (orthogonal polarization channels), as well as the total 1,064 nm return signal (Winker et al., 2007). The depolarization measurements of CALIOP make it possible to distinguish between spherical and non-spherical aerosols (Sassen, 1991), and the signal strength also being influenced by aerosol particle size. Due to the differences in morphology (aspect ratio) and size of the various components of PSC particles, CALIOP can classify PSC particles. Based on this principle, Pitts et al. (2007) developed a PSC classification algorithm using CALIOP data, which has been continuously refined and optimized in subsequent years (Pitts et al., 2009, 2011, 2013, 2018).

In this study, we use the latest CALIPSO Level 2 Polar Stratospheric Cloud Mask V2.00 product (Pitts et al., 2018) from 2006 to 2021, which includes five types of PSCs (STS, NAT-mix, Ice, NAT-enhanced, and wave ice) along CALIPSO orbit tracks and has been widely used to study the characteristics of PSCs and to compare them with model simulations to improve the representation of PSCs in global models (e.g. Tritscher et al., 2021; Zhao et al., 2023; Li et al., 2024). The data spans altitudes from 8.4 km to 30 km, with a vertical resolution of 180 m and a horizontal resolution of 5 km. Benefiting from CALIPSO's polar orbit, the number of observations in the polar regions is significantly higher than in the mid- and low-latitude regions.

#### 2.1.2 Aura MLS measurements

The Microwave Limb Sounder (MLS), on board the National Aeronautics and Space Administration (NASA) Aura satellite launched on July 15, 2004, uses microwave limb sounding technology to provide information from the upper troposphere to the mesosphere (Waters et al., 2006). Aura MLS takes measurements approximately every 25 seconds, recording atmospheric parameters such as temperature and atmospheric constituents from the surface to 90 km. It generates about 3,500 scans per



day, covering latitudes from 82°S to 82°N (Livesey et al., 2006; Read et al., 2006, Livesey et al., 2021). In this study, we use MLS V5.0, Level 3 $H_2O$ and $HNO_3$ from 2004 to 2022. $H_2O$ is retrieved using 190 GHz radiation, with the recommended vertical range spanning from 316 to 0.00215 hPa and an accuracy between 6 % and 22 % in the stratosphere. $HNO_3$ is retrieved using 190 GHz (below 22 hPa) and 240 GHz (above or equal to 22 hPa) radiation, with a recommended vertical range of 215

to 1.5 hPa and an uncertainty range of approximately 0.1 to 2.2 ppbv. V5 data, as recommended by the MLS science team, show improved accuracy over V4, particularly in reducing $H_2O$ drift by about 2 % ~ 4 % per decade (Livesey et al., 2021). Additionally, we utilize daily level 3 data from MLS V5, which are gridded datasets generated by applying a simple "binning" method to the level 2 orbital data.

### 2.1.3 Reanalysis data

The European Centre for Medium-Range Weather Forecasts (ECMWF) ERA5 reanalysis data for temperature and wind from 1979 to 2022 are used in this study, with a spatial resolution of 1° latitude × 1° longitude and 37 levels from 1000 hPa to 1 hPa (Hersbach et al., 2020). The QBO index and the BD circulation are calculated using this dataset. Consistent with previous studies (Zhang et al., 2021), the QBO index in this study is defined as the standardized zonal mean wind between 10°S and 10°N at 50 hPa. The WQBO phase is defined as the QBO index in December greater than 0.5, and the EQBO phase is defined

as less than -0.5. The dataset from 1979 to 2022 includes 14 EQBO phases from December to March and 18 WQBO phases (see Table 1). "El" and "La" in Table 1 refer to the years with strong El Niño and La Niña events, respectively, defined by an Oceanic Niño Index (ONI, https://origin.cpc.ncep.noaa.gov/products/analysis_monitoring/ensostuff/ONI_v5.php) (Glantz and Ramirez, 2020) greater than 1°C or less than -1°C during winter.

**Table 1. List of EQBO and WQBO years from December to March during the study period (1979–2022).**

| EQBO years | WQBO years |
| --- | --- |
| 1979/80, 1981/82, 1984/85[La], 1989/90, 1994/95[El], 1996/97, 1998/99[La], 2001/02, 2003/04, 2005/06, 2007/08[La], 2012/13, 2014/15, 2018/19 | 1982/83[El], 1985/86, 1987/88, 1988/89[La], 1990/91, 1993/94, 1995/96, 1997/98[El], 1999/00[La], 2002/03, 2004/05, 2006/07, 2008/09, 2010/11[La], 2013/14, 2015/16[El], 2016/17, 2020/21[La] |

### 2.2 SLIMCAT Model

TOMCAT/SLIMCAT (hereafter SLIMCAT) is an offline three-dimensional chemical transport model (CTM). It is forced by the ECMWF ERA5 winds and temperatures, which are interpolated to the model grid with a horizontal resolution of 2.8° × 2.8° and a total of 32 levels from the surface to ~60 km (Feng et al., 2021). The model contains a detailed description of

stratospheric chemistry, including heterogeneous reactions on sulfate aerosol and PSC surfaces. It has been widely used in studies of the dynamical transport and chemical reactions in the stratosphere (Dhomse et al., 2011; Chipperfield et al., 2017;



Weber et al., 2021) and has been shown accurately simulate key stratospheric chemical species (Feng et al., 2021; Li et al., 2022). The standard version of the model uses a simplified thermodynamic equilibrium PSC scheme, which includes the parameterization of liquid sulfate aerosols (LA, including binary sulfate aerosols and supercooled ternary solution, STS), solid nitric acid trihydrate (NAT), and ice. The model calculates the equilibrium vapor pressure of $H_2SO_4$ using the expression from Ayers et al. (1980), and when the $H_2SO_4$ concentration exceeds the equilibrium vapour pressure, sulfate aerosol is considered to be present. Notably, the stratospheric $H_2SO_4$ in the model is not provided by initial conditions but is instead calculated based on an assumed number density (10 cm$^{-3}$) of sulfate aerosol, with the stratospheric aerosol surface density (SAD) derived from ftp://iacftp.ethz.ch/pub_read/luo/CMIP6 (Arfeuille et al., 2013; Dhomse et al., 2015; Feng et al., 2021). This scheme also calculates the concentrations, mass fractions, and solubilities of $H_2SO_4$, $HNO_3$, HCl, HBr, HOCl, and HOBr in liquid aerosols (Carslaw et al., 1995a, b).

The presence of NAT and ice is determined based on $HNO_3$, $H_2O$, and temperature using expressions from Hanson and Mauersberger (1988). NAT is considered to be present when the concentration of $HNO_3$ exceeds 10 times its equilibrium vapour pressure (Grooß et al., 2018). The model assumes that NAT particles exist in two modal radii (0.5 μm and 6.5 μm), and the SAD required for heterogeneous reactions is derived from small-radius NAT particles calculated by the condensable amount of $HNO_3$, with a limit of 1 cm$^{-3}$ for the number density of small particles. The remaining $HNO_3$ condenses into large-radius NAT particles (Davies et al., 2002; Feng et al., 2005). Ice is assumed to form when $H_2O$ concentration exceeds its equilibrium vapour pressure, and the model assumes a number density of 10 cm$^{-3}$ for ice particles, with their SAD calculated from the condensable amount of $H_2O$. The model also uses simplified denitrification and dehydration schemes, with large-radius NAT particles and ice particles assumed to settle at velocities of 1100 m/day and 1500 m/day, respectively (Feng et al., 2011), to redistribute $HNO_3$ and $H_2O$.

In this study, we use SLIMCAT to simulate the period 1979–2022. In addition, fixed ozone-depleting substances (ODS) for the year 2000 and climatologies of sulfate aerosol SAD and solar flux were used to exclude the effects of variations in sulfate aerosol SAD, solar flux, and anthropogenic ODS emissions. Based on the Arctic winter and early spring $HNO_3$ and $H_2O$ derived from SLIMCAT, we diagnosed offline the equilibrium temperature of NAT ($T_{NAT}$) according to Hanson and Mauersberger (1988), which is the warmest temperature at which PSC particles can theoretically start to form and persist (Tritscher et al., 2021; Leroux and Noel, 2024). We consider PSCs to be present when the temperature at the grid point falls below $T_{NAT}$ and calculate the PSC coverage areas. These areas based on $T_{NAT}$ are the theoretical maximum areas of PSC. Li et al. (2024) have shown that the spatial and temporal variability characteristics of the SLIMCAT PSC areas calculated by this method are in good agreement with those observed by CALIPSO.

## 2.3 PSC area calculation

In this study, two different methods are used to calculate the PSC coverage areas of CALIPSO and SLIMCAT, respectively. For SLIMCAT, the PSC coverage area is computed by summing the areas of model grid cells containing PSCs, hereafter referred to as the "Grid method". Since CALIPSO observations of PSCs along its orbit do not cover all model grid cells, the





CALIPSO PSC area is calculated using a statistical method referred to Pitts et al. (2018) (hereafter, the "P18 method"), which divides 50°–90°N into 10 latitude bands, and the PSC area is estimated as the sum of the occurrence frequency of PSC in the 10 latitude bands multiplied by the area of each band. Li et al. (2024) compared the differences in SLIMCAT PSC areas calculated using the "Grid method" and the "P18 method". They indicated that the areas derived from both methods are similar. However, when the "P18 method" is used, the daily variability in the SLIMCAT PSC area increases slightly. To avoid this

issue, the "P18 method" is used to calculate the PSC coverage areas of CALIPSO, while the "Grid method" is applied to SLIMCAT.

## 3 Results

Figure 1 (left) shows the relationship between the Arctic PSC area in January and the QBO index of the previous month (December). Both CALIPSO observations and SLIMCAT simulations show that the PSC area increases with increasing QBO

index. However, the CALIPSO PSC areas do not show a statistically significant correlation with the QBO index ($p = 0.29$), likely due at least in part to the limited sample size of the satellite observations. In contrast, the SLIMCAT PSC area shows greater variation with the QBO index, and the regression line (red line) passes the significance test ($p = 0.004$), likely due to the larger sample size. In addition to the correlation, the probability distribution of the PSC area also shows a distinct difference between the WQBO and EQBO phases. During the EQBO phase, a large portion of the PSC area is near zero. For example, of

the 14 SLIMCAT samples, 6 samples have values less than 1 million km², while 3 out of 4 CALIPSO samples are close to zero. In contrast, during the WQBO phase, all SLIMCAT samples exceed 1 million km², and only two CALIPSO samples fall below this threshold. The probability distribution functions (PDF) of the PSC area during the WQBO and EQBO phases also support the above conclusions (right panel of Fig. 1). Compared to the WQBO, the PSC area during the EQBO phase shifts to smaller values, suggesting that the tropical QBO can influence Arctic PSC areal extent in winter. In addition, during the WQBO

phase, the PDF of the PSC area is closer to a uniform distribution, in contrast to the skewed distribution during the EQBO phase. Note that large PSC areas can occur during the EQBO phase, because the Arctic stratospheric vortex is influenced not only by the QBO but also by other factors, such as ENSO (Brönnimann et al., 2004; Garfinkel and Hartmann, 2008; Zhang et al., 2022).



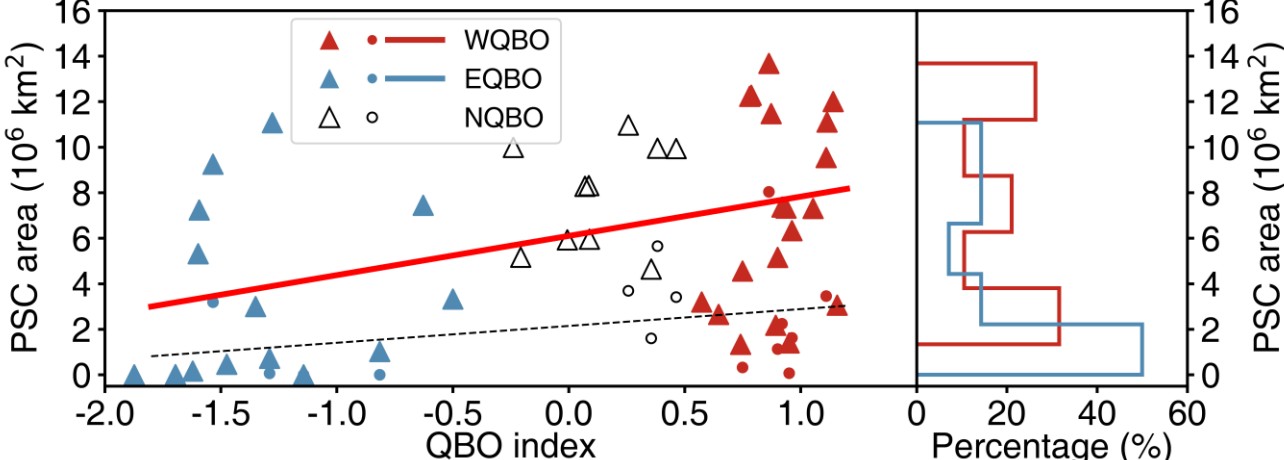

**Figure 1. Arctic PSC area on 500 K isentropic level in January plotted against the QBO index in December (left). Triangles represent the PSC area simulated by SLIMCAT from 1980 to 2022, while circles represent the PSC area observed by CALIPSO from 2007 to 2021. Blue markers and red markers represent the PSC area during EQBO and WQBO, respectively. The red and black lines show the linear regression of the QBO index and the PSC area for SLIMCAT and CALIPSO, respectively. The solid line is statistically significant at the 95% confidence level, while the dashed line is not. The probability distribution functions (PDF) of the PSC area for the two QBO phases are derived from the SLIMCAT simulation (right).**

To investigate the relationship between the QBO and PSCs, composite analyses of PSCs are performed. It is important to note that before performing the composite analysis on all variables, the linear trends of the variables are removed. However, our results indicate that the detrending has a minimal impact on the composite results. Figure 2 shows the differences in the time series of the Arctic PSC area during winter derived from the CALIPSO observations and the SLIMCAT simulations between the WQBO and EQBO phases. The CALIPSO observations show that during the WQBO phase, the PSC area in January and February on 400–600 K isentropic level is significantly larger than during the EQBO phase, whereas there is a non-significant negative area anomaly in mid-December. The differences derived from SLIMCAT for the same periods are shown in Fig. 2b. Although the differences in the SLIMCAT PSC area are larger than those observed by CALIPSO, the simulations successfully reproduce the temporal and spatial distribution of the anomalies, such as the negative area anomaly in mid-December and the maximum positive anomaly on ~500 K in January. We then use a longer dataset derived from SLIMCAT simulations from 1979−2022 to validate this conclusion (Fig. 2c). With increasing sample size, significant regions markedly increase. Furthermore, the negative area anomaly in mid-December shifts to a positive anomaly, although these positive anomalies are not significant, possibly due to the influence of negative anomalies in the 2006−2021 period. In contrast, the significance of the December monthly mean anomalies increases, as can be seen in the right panel of Fig. 2c. Furthermore, the maximum positive anomaly appears in January on ~500 K isentropic level, and the height of the maximum anomaly gradually decreases from December to February, which is consistent with the climatological characteristics of the PSC area (Pitts et al., 2018; Li et al., 2024). These results confirm that the December QBO phase can significantly influence the PSC area in winter, with the PSC area being significantly larger during the WQBO phase compared to the EQBO phase.



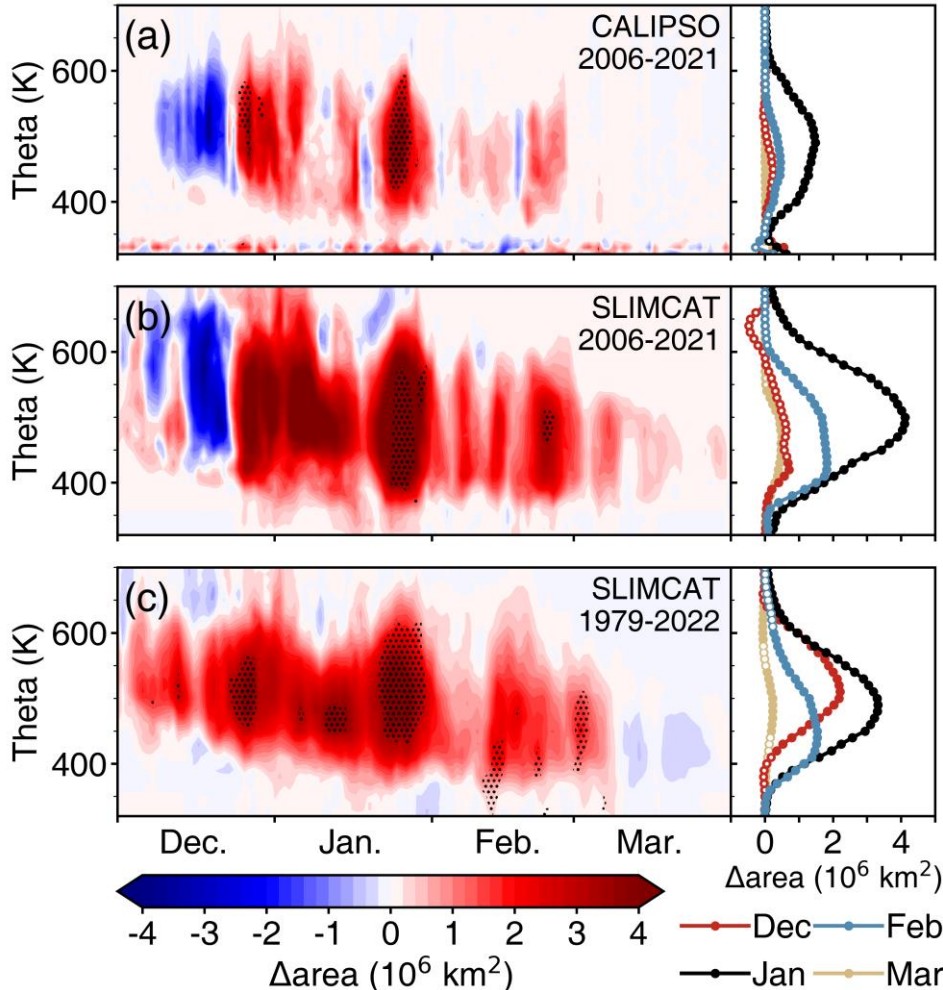

**Figure 2. Differences in Arctic PSC area between the WQBO and EQBO phases, with variation overtime on the left panel and monthly averages on the right panel, derived from (a) CALIPSO observations from 2006−2021, and SLIMCAT simulations from (b) 2006−2021 and (c) 1979−2022. Black dotted regions in the left panels and solid filled symbols in the right panels indicate the differences in PSC area are statistically significant at the 95% confidence level according to the Student's t-test.**

We bin the CALIPSO PSC data into the SLIMCAT model grid, where the monthly mean CALIPSO PSC occurrence frequency is defined as the number of PSC observations in each month per grid point divided by the total number of observations. The SLIMCAT PSC occurrence frequency is defined as the number of PSC occurrences at a grid point within a month divided by the number of days in that month. Figure 3 presents the spatial distribution of the differences in PSC occurrence frequency on 500 K derived from the CALIPSO observations. During the WQBO phase, the Arctic PSC occurrence frequency shows an overall positive anomaly in winter (Fig. 3a), with a clear zonal asymmetry. The largest positive anomaly occurs over Greenland and the Barents Sea. The anomaly is largest in January, followed by December, and smallest in February. The spatial distribution of this anomaly may be related to the climatology of PSC occurrence frequency. According to Pitts et





al. (2018), the monthly mean PSC occurrence frequency in the Arctic peaks in January, with a clear zonal asymmetry, and the highest frequency is centered over the Barents Sea, corresponding to the lowest geopotential height of the Arctic stratospheric vortex (Zhang et al., 2016). The zonal asymmetry of PSC occurrence frequency anomalies in Fig. 3 may also be influenced by

255 QBO-induced changes in the polar vortex position, which we discuss further below. In addition, due to the smaller sample size, especially at lower latitudes, banded areas of positive and negative anomalies similar to the CALIPSO satellite orbit are exhibited.

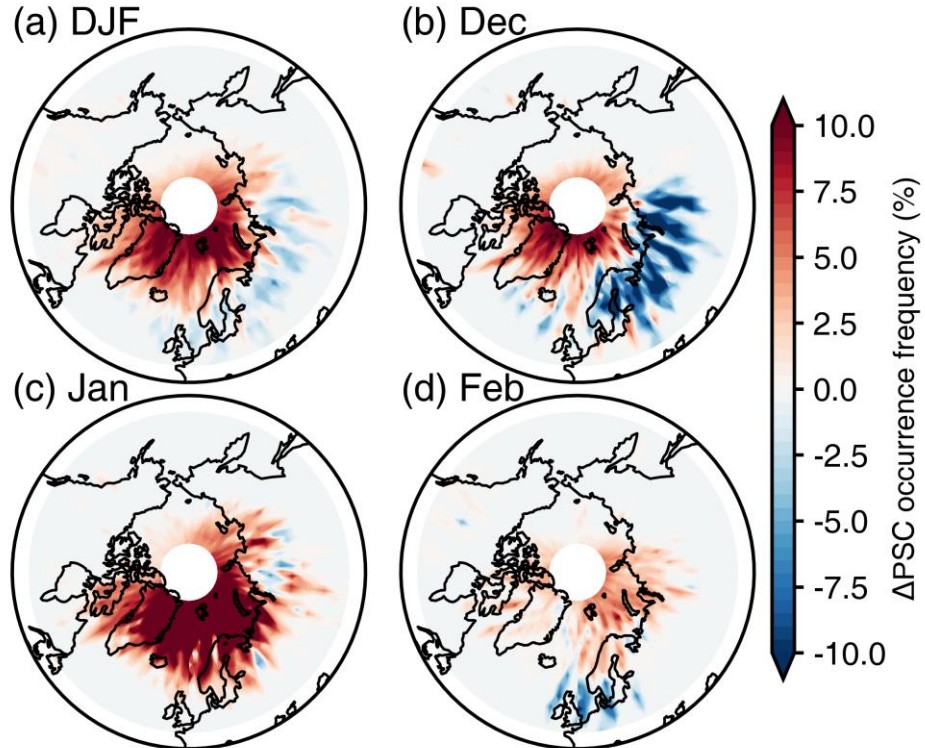

**Figure 3. Differences in PSC occurrence frequency between the WQBO and EQBO phases on the 500 K isentropic level derived**
**from the CALIPSO observations for the period 2006–2021 for (a) winter average (December to February), (b) December, (c) January, and (d) February. Black dotted regions indicate the differences in PSC occurrence frequency are statistically significant at the 95% confidence level according to the Student's *t*-test.**

Figure 4 presents the spatial distribution of the differences in the PSC occurrence frequency between the WQBO and EQBO phases from 1979 to 2022, as simulated by SLIMCAT. Compared to the CALIPSO results (Fig. 3), the increased
sample size from SLIMCAT simulations leads to a larger anomalous region being statistically significant. In addition, the differences in SLIMCAT PSC occurrence frequency between the WQBO and EQBO phases are greater than that in CALIPSO PSC. Similar to the CALIPSO results, SLIMCAT simulations show notable positive anomalies during the WQBO phase compared to the EQBO phase, which also shows a zonal asymmetric structure. Furthermore, the zonal asymmetry appears to be more pronounced in December and February than in January. The non-significant negative anomalies over northern Eurasia





are observed in December and February (Fig. 4b and d), which is also present in the CALIPSO observations (Fig. 3b). Zhang et al. (2019) suggested that during the WQBO phase, the NH polar vortex tends to shift towards North America and away from Eurasia during winter compared to the EQBO phase, which may affect the distribution of PSCs, leading to a decrease in PSCs over Eurasia. However, this negative anomaly does not occur in January, probably due to a less pronounced polar vortex shift associated with the QBO. Overall, during the WQBO phase, the Arctic winter shows a positive anomaly in PSC occurrence frequency, the centre of which is located between Greenland and the Barents Sea. Compared to the climatological maximum frequency of PSC occurrence (Li et al., 2024), this positive anomaly lies upstream (west) of the maximum occurrence frequency.

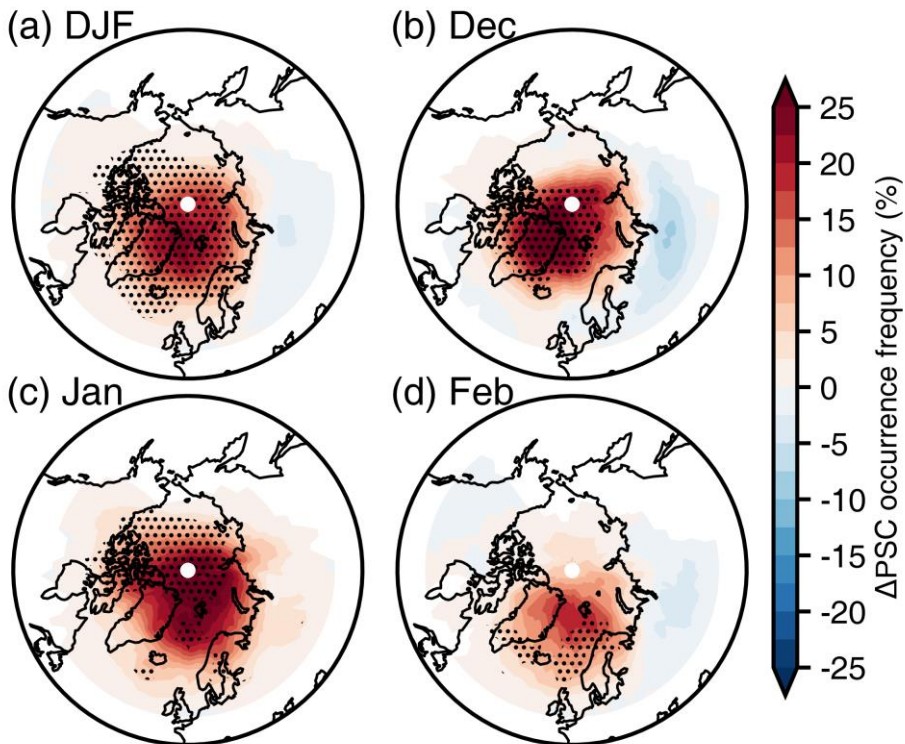

**Figure 4. Same as Fig. 3 but derived from the SLIMCAT simulations for the period 1979–2022.**

It is known that ENSO is another important factor influencing the Arctic stratospheric vortex in winter. During El Niño events, the Aleutian Low is deepened, planetary wave activity intensifies, and more planetary waves propagate into the mid- and high-latitudes of the NH stratosphere, ultimately making the Arctic stratospheric polar vortex weaker and warmer than during La Niña events (Garfinkel and Hartmann, 2008; Salminen et al., 2020). Therefore, we should examine whether the conclusions that the QBO affects the PSC still hold after excluding the ENSO years. Figure 5 shows the differences in PSC occurrence frequency between the WQBO and EQBO phases, excluding years with strong ENSO. The results are similar to



those in Fig. 4, thus confirming the validity of the previous conclusions. However, excluding strong ENSO years reduces the sample size for composite analysis, resulting in a smaller extent of statistically significant regions compared to Fig. 4.

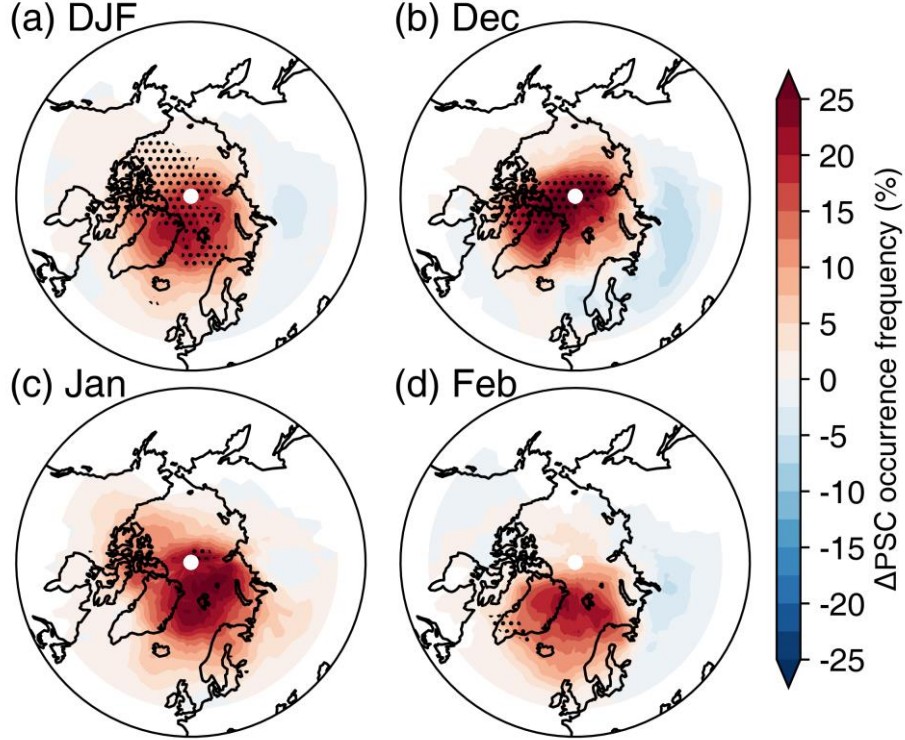

**Figure 5. Differences in PSC occurrence frequency between WQBO and EQBO phases on the 500 K isentropic level (excluding strong ENSO events) from SLIMCAT simulations for (a) winter average (December to February), (b) December, (c) January, and (d) February during 1979–2022. Black dotted regions indicate that the differences in PSC occurrence frequency are statistically significant at the 95% confidence level according to the Student's *t*-test.**

Figure 6 shows the zonal mean differences in PSC occurrence frequency during the WQBO and EQBO phases derived from the SLIMCAT simulations. Consistent with the result from Zhang et al. (2021), which reports differences in the number of days with temperatures below 195 K, PSC occurrence frequencies are higher during the WQBO phase compared to the EQBO phase, especially in December and January. Furthermore, the seasonal variation of the differences in PSC occurrence frequency is consistent with the seasonal variation in PSC area differences shown in Fig. 1, with a gradual decrease in the altitude at which the maximum difference is located from December to February. Overall, in Fig. 6a, the differences in PSC occurrence frequency peaks near 500 K, with positive anomalies exceeding 10% north of 75°N.





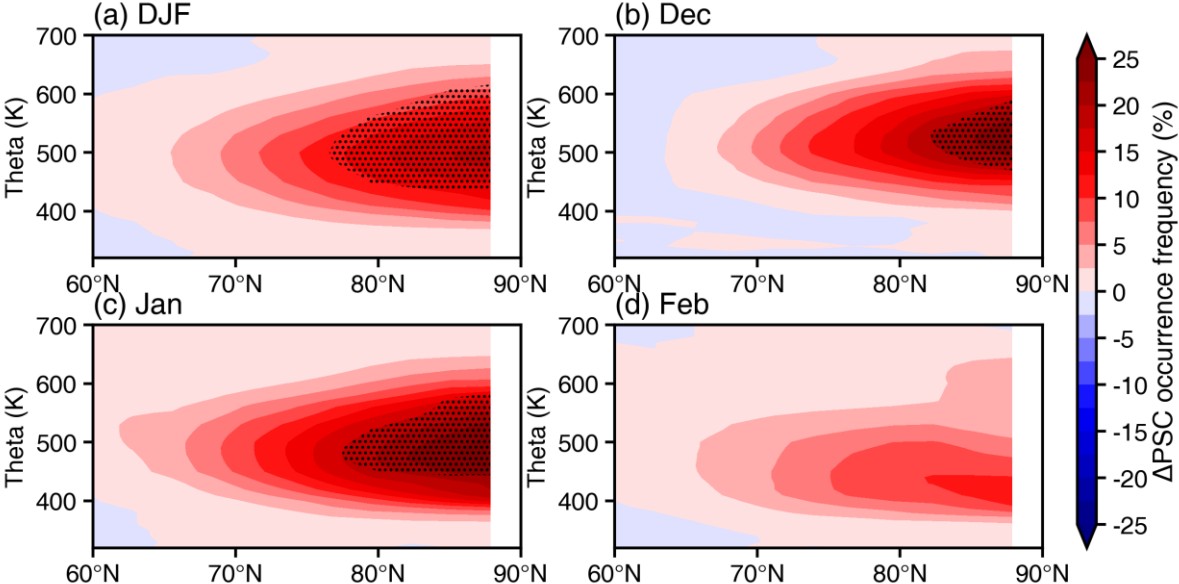

**Figure 6. Differences in zonal-mean PSC occurrence frequency between WQBO and EQBO derived from the SLIMCAT simulations for the period 1979–2022 for (a) winter average (December to February), (b) December, (c) January, and (d) February. Black dotted regions indicate that the differences in PSC occurrence frequency are statistically significant at the 95% confidence level according to the Student's *t*-test.**

The above analyses reveal that the tropical QBO can significantly influence the Arctic PSC occurrence. During the WQBO phase, the Arctic PSCs have a larger coverage area and higher occurrence frequency, which may contribute to an increase in Arctic stratospheric ozone depletion. In addition, as mentioned in the introduction, the QBO may influence Arctic PSC formation through modulating temperature, $H_2O$, and $HNO_3$. However, it is unclear how QBO-induced changes in these factors affect PSC. The underlying mechanisms by which the QBO influences Arctic PSC are now discussed.

Figure 7 presents the climatological temperature and the QBO composite differences in temperature. The climatological temperature exhibits a clear zonal asymmetry, with the lowest temperatures biased towards the Eurasian continent, especially near the Barents Sea. This asymmetry is mainly driven by the zonal wavenumber-1 activity, where the climatological polar vortex is shifted towards Eurasia (Zhang et al., 2016; Wang et al., 2022). Significant negative temperature anomalies are observed in most of Arctic regions between the WQBO and EQBO phases, consistent with previous studies (Holton and Tan, 1980, 1982; Zhang et al., 2019). Furthermore, the temperature anomalies associated with the QBO show a distinct zonal asymmetry, with the centre of the anomalies biased towards North America. During the WQBO phase, the polar vortex is shifted towards North America, contributing to this asymmetry (Zhang et al., 2019). It is important to note that the asymmetry of the negative temperature anomaly is more pronounced in December and February, while it is relatively weaker in January.




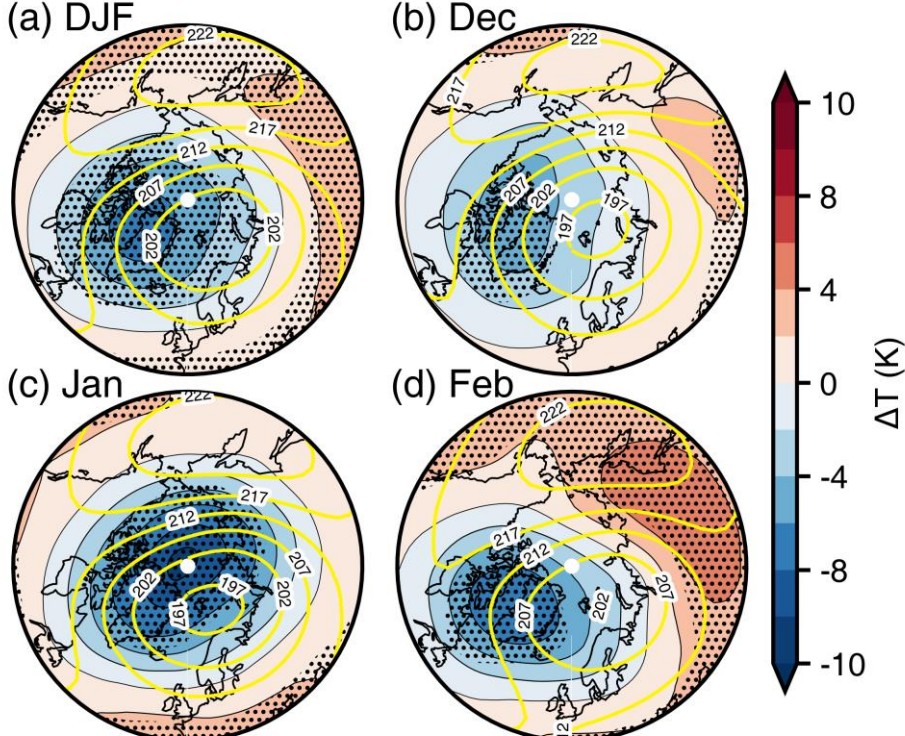

 **Figure. 7 Climatological temperature (yellow contours) and the differences in temperature between WQBO and EQBO phases on the 500 K isentropic level for the period 1979–2022 (shadings, WQBO–EQBO) derived from ERA5 data for (a) winter average (December to February), (b) December, (c) January, and (d) February. Black dotted regions indicate the differences in temperature are statistically significant at the 95% confidence level according to the Student's *t*-test.**

However, the negative temperature anomalies do not coincide well with the positive PSC occurrence anomalies, with the

negative temperature anomalies located to the west of the positive PSC occurrence anomalies (Fig. 4). This is because PSC

formation requires the ambient temperature to be sufficiently lower than the existence temperature (about 195 K; Tritscher et

al., 2021). Only the QBO-induced Arctic cooling, occurring over the climatological cold centre, could lead to increased PSC

occurrence. Therefore, the positive PSC anomalies are located near the climatological cold centre and appear to the east of the

negative temperature anomalies.

In addition to the temperature, sufficient concentrations of $H_2O$ and $HNO_3$ are critical for PSC formation (Li et al., 2024).

Figure 8 shows the differences in $H_2O$ and $HNO_3$ concentrations between the WQBO and EQBO phases observed by MLS

and simulated by SLIMCAT over the Arctic (60°N–82°N). Both the MLS observations (Fig. 8a) and the SLIMCAT simulation

(Fig. 8c) show positive $H_2O$ anomalies above the 450 K isentropic level, with the maximum anomaly occurring in January and

approaching ~0.3 ppmv on the 650 K isentropic level. In addition, the altitude of the maximum positive anomaly of $H_2O$

decreases with time similar to the decrease in the maximum positive anomaly of the PSC area, which may be related to the

downward shift of the coldest centre of the polar vortex with time (Coy et al., 1997; Lawrence et al., 2018).



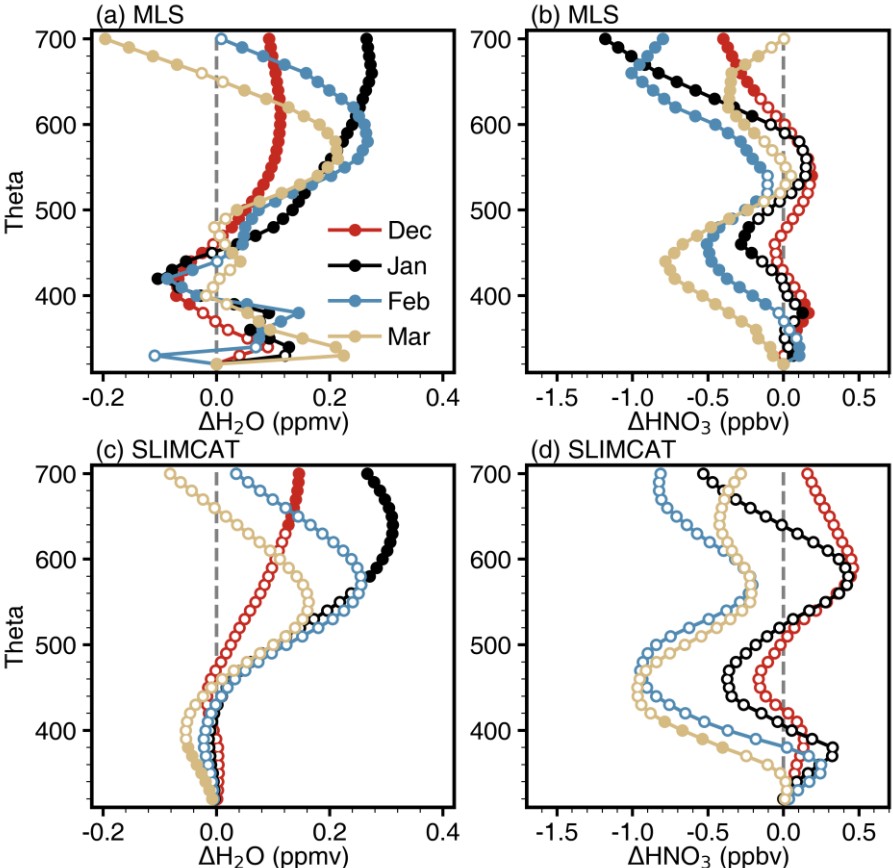

**Figure 8. Differences in H₂O and HNO₃ averaged over 60°−82°N between the WQBO and EQBO phases from (a, b) MLS observations and (c, d) SLIMCAT simulations for the period 2004−2021. Solid filled symbols indicate the differences are statistically significant at the 95% confidence level according to the Student's t-test.**

MLS (Fig. 8b) and SLIMCAT (Fig. 8d) both show similar trends in the $HNO_3$ anomalies with height for all months: first decreasing, then increasing, and then decreasing again. However, MLS observations show negative $HNO_3$ anomalies above 550 K from December to March, whereas SLIMCAT simulations show positive anomalies in December and January. In addition, SLIMCAT well reproduces the negative anomaly $HNO_3$ on 400–500 K, while the negative anomaly is smaller in December and January than in February and March.

Stratospheric chemical species are influenced by both dynamical and chemical processes. Due to the different chemical and physical properties of the different species, the factors affecting their concentrations also vary. Stratospheric $H_2O$ has two primary sources: transport from the troposphere and the oxidation of methane (Texier et al., 1988). Since stratospheric temperatures increase with altitude, $H_2O$ produced by $CH_4$ oxidation reactions (primarily $CH_4 + OH \rightarrow CH_3 + H_2O$) is more important in the older air in the upper stratosphere. In summer and autumn, $H_2O$ produced by chemical reactions accounts for 50% of the total stratospheric $H_2O$ at the stratopause. In contrast, in the lower stratosphere, where temperatures are lower, $CH_4$



oxidation is weaker, and $H_2O$ mainly originates from dynamical transport (Thölix et al., 2016). Therefore, chemical reactions are not the primary cause of the differences in $H_2O$ between the WQBO and EQBO phases.

Hansen et al. (2013) provided an explanation for the positive $H_2O$ anomalies during the WQBO phase, attributing these anomalies to dynamical transport processes. Specifically, they proposed that the vertical transport anomaly of the BD circulation induced by the QBO leads to an increase in $H_2O$. To explore the relationship between $H_2O$ anomalies and BD circulation anomalies, Figure 9 shows the time evolution of the $H_2O$ anomalies and climatology and the vertical velocity anomalies of the BD circulation in the Arctic lower stratosphere. The $H_2O$ concentration in the polar stratosphere increases with altitude, which is caused by the $CH_4$ oxidation (Grooß and Russell, 2005). If the BD circulation anomalies are the direct

cause of the $H_2O$ anomalies, then during the upward anomaly of the BD circulation, it would transport low $H_2O$ air from the lower layers to the upper layers, resulting in a negative $H_2O$ anomaly. However, we note that regardless of whether the BD circulation anomalies are upward or downward, the $H_2O$ anomalies remain positive. Therefore, the BD circulation anomalies cannot account for the $H_2O$ anomalies.

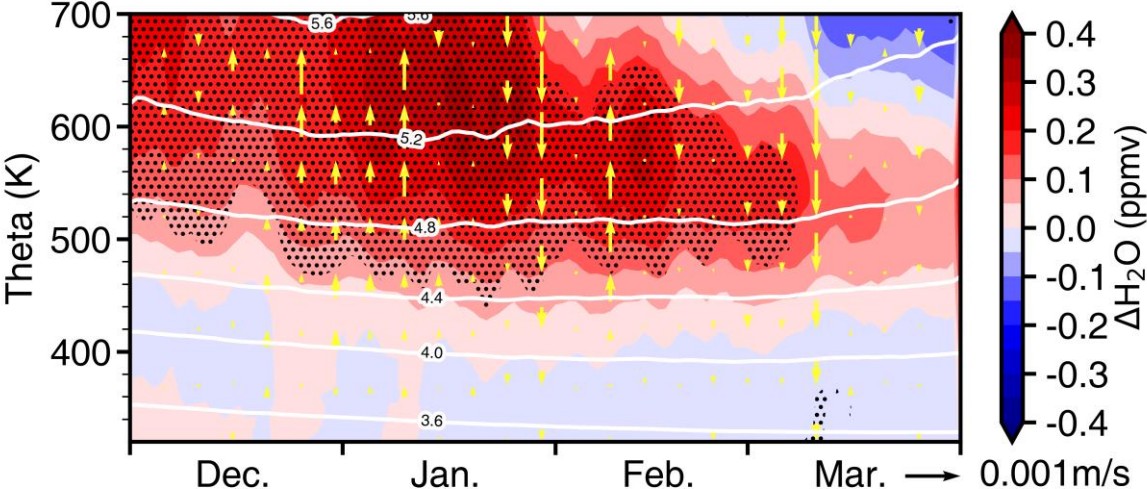

**Figure 9. Differences in $H_2O$ (shading) and vertical component of BD circulation (yellow arrows) between the WQBO and EQBO phases from SLIMCAT simulations for the period 1979−2022. The white contours are the climatological concentration of the $H_2O$. Black dotted regions indicate the differences in $H_2O$ are statistically significant at the 95% confidence level according to the Student's t-test.**

Another potential cause for the anomalies in Arctic stratospheric chemical species could be the stratospheric polar vortex.

This isolates the polar air from that outside the vortex. Khosrawi et al. (2016) found a clear inverse correlation between $H_2O$ and temperature in the Arctic lower stratosphere, suggesting a high $H_2O$ concentration in the Arctic lower stratosphere under a strong polar vortex. Pan et al. (2002) found that decreased amounts of tracers can be transported outside the vortex during strong polar vortex conditions compared to weak polar vortex conditions. In the lower stratosphere, $H_2O$ concentrations are higher in the polar regions than in the mid-latitudes at the same altitude (Randel et al., 2001; Jiang et al., 2015), which is related

to the downwelling branch of the BD circulation in the polar regions and the upwelling branch in the tropics (Morrey and




Harwood, 1998). Therefore, a strong polar vortex may prevent the transport of high moisture air at high latitudes to mid-latitudes, leading to increased $H_2O$ concentrations within the vortex. Here, the zonal winds averaged over 60°N–65°N are used to define the strength of the Arctic polar vortex. The relationship between polar vortex strength and $H_2O$ concentration across different latitude bands is shown in Figure 10a and c. Given the overall increasing trend in stratospheric $H_2O$ concentration

driven by global warming and increased methane emissions (Nedoluha et al., 1998; Oltmans et al., 2000; Dessler et al., 2013; Smalley et al., 2017), the monthly mean $H_2O$ have been detrended before analysis. Both the MLS observation and the SLIMCAT simulation reveal a strong positive correlation between the Arctic $H_2O$ concentration and the polar vortex strength, with correlation coefficients of 0.84 for the observations and 0.74 for the simulation. In contrast, the $H_2O$ concentration in the mid-latitudes decreases as the polar vortex strengthens. The relatively slow decrease of $H_2O$ in the mid-latitudes can be

explained by the larger volume of air at the same latitude range, which makes it less susceptible to the influence of polar $H_2O$ transport. The opposite evolution of $H_2O$ inside and outside the polar vortex suggests that a stronger vortex suppresses the transport of $H_2O$ from the polar region to mid-latitudes. Therefore, a stronger polar vortex induced by the WQBO isolates high concentrations of $H_2O$ within the polar region, resulting in higher $H_2O$ concentrations compared to the EQBO.

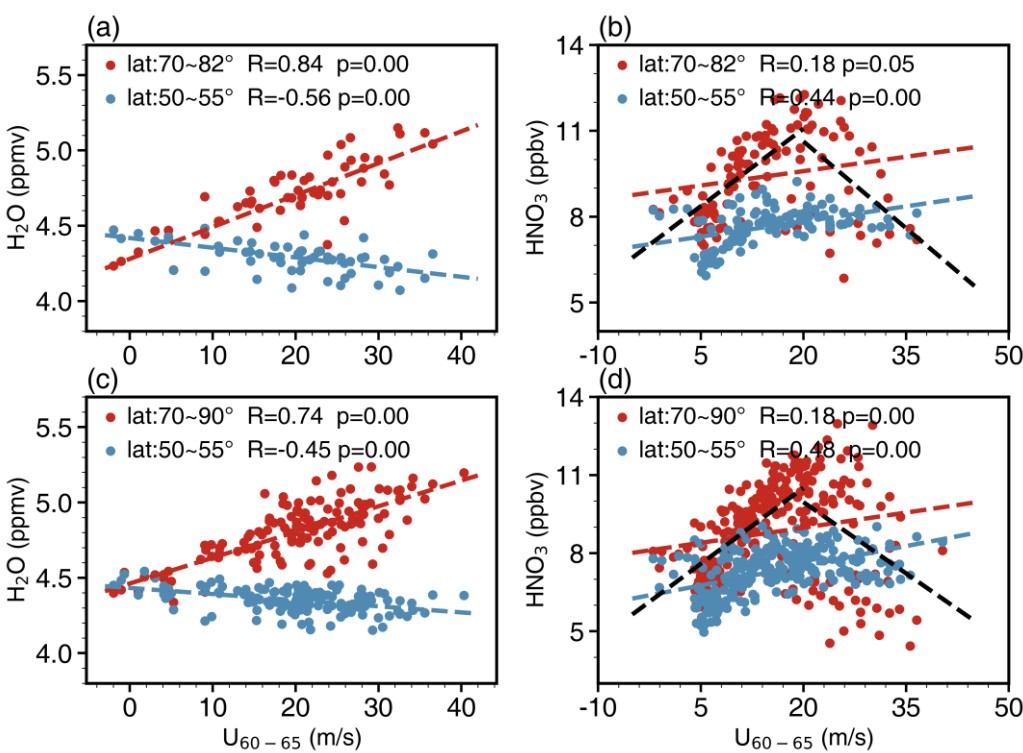

**Figure 10. Relationship between (a, c) $H_2O$ and (b, d) $HNO_3$ and the zonal mean wind between 60°N and 65°N on the 500 K isentropic level. Data from (a, b) MLS observations (2004–2022) and (c, d) SLIMCAT simulation (1979–2022) for the period (a, c) December to February and (b, d) September to February. Red represents high-latitude regions (MLS: 70°N–82°N, SLIMCAT: 70°N–90°N), and blue represents mid-latitude regions (50°N–55°N). Dashed lines represent linear fits, with R denoting the correlation coefficient. The black dashed line provides a segmented fit before and after 20 m/s at high latitudes. Values of p < 0.05 are considered statistically**

**significant.**





Since the change in stratospheric $HNO_3$ from December to February is relatively small, we display the relationship between $HNO_3$ from MLS and SLIMCAT and polar vortex strength from September to February (Fig. 10b and d). Overall, $HNO_3$ concentrations in the mid- and high-latitudes of the NH stratosphere increase with polar vortex strength. The correlation between $HNO_3$ and polar vortex strength is stronger in the mid-latitudes (with coefficients of 0.44 for observations and 0.48

for simulations) and is statistically significant, while the correlation is weaker in the high-latitudes (coefficients of 0.18 for observations and simulations), with weaker significance. Notably, the relationship between $HNO_3$ concentrations and polar vortex strength in the high-latitude stratosphere shows two distinct trends around wind speeds of 20 m/s. When wind speeds are below 20 m/s, $HNO_3$ concentrations increase with wind speed, but above 20 m/s, $HNO_3$ concentrations decrease.

$HNO_3$ is an important reservoir molecule for stratospheric reactive nitrogen ($NO_y$), produced by the oxidation of $NO_x$

($NO_2 + OH + M \rightarrow HNO_3 + M$). However, under solar radiation, $HNO_3$ is destroyed through photolysis ($HNO_3 + h\nu \rightarrow OH + NO_2$) and reactions with OH ($HNO_3 + OH \rightarrow H_2O + NO_3$) (Keys et al., 1993), leading to lower concentrations in the Arctic stratosphere in summer. As solar radiation decreases in autumn, the stratospheric polar vortex begins to form, and the reduction in photolysis and heterogeneous chemical reactions on background aerosols causes $HNO_3$ concentrations to increase gradually (Solomon and Keys, 1992; Keys et al., 1993). Therefore, as strength of the polar vortex increases, $HNO_3$ concentrations

increases in the mid- and high-latitude of the NH stratosphere. However, once wind speeds reach a certain level and the polar vortex becomes sufficiently strong and isolated, temperatures can become low enough for NAT to form. If a sufficient amount of NAT has formed, denitrification occurs within the polar vortex due to the NAT sedimentation, leading to a decrease in $HNO_3$ concentrations. This explains why the observed decrease in $HNO_3$ concentrations above wind speeds of 20 m/s in Fig. 10b and d. During winter and early spring, Arctic temperatures are sufficiently low for denitrification to occur. Therefore, the

lower temperatures during the WQBO phase result in lower $HNO_3$ concentrations compared to the EQBO phase (Fig. 8b and d).

Both $H_2O$ and $HNO_3$ concentrations are influenced by PSC sedimentation. When the polar vortex is strong enough, both $H_2O$ and $HNO_3$ concentrations decrease. It is important to note that different types of PSCs have different effects on stratospheric chemicals. NAT sedimentation primarily affects $HNO_3$ concentrations, while ice PSC sedimentation primarily

affects $H_2O$ concentrations. Due to distinct temperature climatology, there is a clear disparity in the composition of Arctic and Antarctic PSCs, with ice PSCs constituting about 25% of Antarctic PSCs but less than 5% of Arctic PSCs (Pitts et al., 2018). As a result, ice PSC-related dehydration occurs annually in the Antarctic but rarely in the Arctic (Khaykin et al., 2013). Since NAT accounts for up to 60% of Arctic PSC, the effects of the QBO on Arctic stratospheric $HNO_3$ and $H_2O$ are markedly different.

As mentioned above, the WQBO phase tends to lower the Arctic temperatures (Fig. 7), reduce $HNO_3$ concentrations, and increase the $H_2O$ concentrations (Fig. 8). The lower temperatures and higher $H_2O$ concentrations are favourable for PSC formation, while the lower $HNO_3$ concentrations are unfavourable (Hanson and Mauersberger, 1988). However, their relative contributions remain unclear. To address this issue, we perform sensitivity analyses in which the temperature, $H_2O$, and $HNO_3$ are adjusted by adding or subtracting 50 % of the differences between the WQBO and EQBO phases. It is important to note




that we did not adjust temperature, $H_2O$, and $HNO_3$ during the model runs, but instead made these adjustments during the offline diagnosis of the PSCs. Our adjustment strategy aims to reduce the PSC area during the WQBO phase and increase it during the EQBO phase, to investigate the main factors influencing Arctic PSCs by changing temperature, $H_2O$, and $HNO_3$. Table 2 provides the descriptions of the six sensitivity analyses.

**Table 2. Description of the sensitivity analyses, where Δ represents the differences between the WQBO and EQBO phases.**

| Name | Change | PSC area change | Description |
|---|---|---|---|
| W_high T | T-50 %×ΔT[1] | Decrease[2] | The temperature during the WQBO phase is subtracted by 50% of the temperature differences. |
| E_low T | T+50 %×ΔT | Increase | The temperature during the EQBO phase is added by 50% of the temperature differences. |
| W_less $H_2O$ | $H_2O$-50 %×Δ$H_2O$ | Decrease | The $H_2O$ during the WQBO phase is subtracted by 50% of the $H_2O$ differences. |
| E_more $H_2O$ | $H_2O$+50 %×Δ$H_2O$ | Increase | The $H_2O$ during the EQBO phase is added by 50% of the $H_2O$ differences. |
| W_less $HNO_3$ | $HNO_3$+50 %×Δ$HNO_3$ | Decrease | The $HNO_3$ during the WQBO phase is added by 50% of the $HNO_3$ differences. |
| E_more $HNO_3$ | $HNO_3$-50 %×Δ$HNO_3$ | Increase | The $HNO_3$ during the EQBO phase is subtracted by 50% of the $HNO_3$ differences. |

Note: 1. According to the above composite analyses, temperature and $HNO_3$ show negative anomalies and $H_2O$ shows positive anomalies during the WQBO phase. 2. Higher temperatures, less $H_2O$ and $HNO_3$ are unfavourable for PSC formation.

Note that the temperature variation induced by the QBO accounts for most of the changes in the PSC areas (Fig. 11a–d). In particular, during the WQBO phase, when the temperature is subtracted by 50 % of the differences between the WQBO and
EQBO phases, that means warmer than the original temperature, the PSC areas decrease and approach those during the EQBO phase. In contrast, during the EQBO phase, when the temperature is added by 50 % of the differences between the WQBO and EQBO phases, that means cooler, the PSC area increases slightly. This suggests that PSC formation is more sensitive to temperature variations during the WQBO phase. Since Arctic temperatures are concentrated around the PSC formation threshold (Fig. 7), even small changes in temperature would result in significant variations in PSC. QBO-induced changes in
$H_2O$ and $HNO_3$ lead to only small changes in PSC. During the WQBO phase, when the $H_2O$ decreases by 50 % of the difference, the PSC area becomes smaller, although the magnitude of the change is relatively small (Fig. 11e–h). QBO-induced $HNO_3$ changes affect the PSC in late winter and early spring (Fig. 11k–l). During the WQBO phase, lower Arctic temperatures lead to stronger denitrification, resulting in less $HNO_3$ participating in PSC formation. These results show that it is important to simulate the NAT sedimentation process in the model accurately.




**Figure 11.** PSC area from December to March during the WQBO (red) and EQBO (blue) phases. The first, second, and third rows show the results from sensitivity analyses with imposed temperature, $H_2O$, and $HNO_3$ changes. Solid lines represent results before the changes, and dashed lines represent results after the changes.



## 4 Discussion and Conclusions

Using CALIPSO observations data and the SLIMCAT simulations, we analyze the impact of the QBO on the occurrence of Arctic PSCs. The results show that as the QBO index increases, the coverage area of Arctic PSCs gradually increases. Compared to the SLIMCAT simulations for the same period (2006–2021), the CALIPSO PSC area anomalies between the WQBO and EQBO are smaller, while SLIMCAT reproduces the distribution of these anomalies accurately. SLIMCAT simulations for the period 1979–2022 show that the winter PSC area is significantly larger during the WQBO phase than during

the EQBO phase, with the maximum positive anomaly occurring near the 500 K isentropic level in January. From December to February, this maximum positive anomaly gradually decreases in altitude. Both CALIPSO observations and SLIMCAT simulations show a clear zonal asymmetry in the PSC occurrence frequency anomaly between the WQBO and EWBO phases on the 500 K isentropic level. During the WQBO phase, there is a positive anomaly in the occurrence frequency of Arctic winter PSCs, with the maximum positive anomaly located between Greenland and the Barents Sea. This region is close to the

region of maximum PSC occurrence frequency (Li et al., 2024) and the location of the lowest geopotential height of the Arctic polar vortex (Zhang et al., 2016).

        The QBO affects Arctic PSCs through its effects on stratospheric temperature, $H_2O$, and $HNO_3$. We have analyzed the temperature, $H_2O$, and $HNO_3$ anomalies induced by the QBO. Compared to the EQBO phase, there is a zonal asymmetry negative temperature anomaly in the Arctic stratosphere during the WQBO phase. The maximum negative anomaly is located

upstream (west) of the climatological minimum temperature. MLS satellite observations and SLIMCAT simulations show a positive $H_2O$ anomaly and a negative $HNO_3$ anomaly in the Arctic lower stratosphere during the WQBO phase. The positive $H_2O$ anomaly is mainly due to the stronger polar vortex during the WQBO phase than during the EQBO phase, which prevents the transport of high-moisture air at high latitudes to the mid-latitudes, causing $H_2O$ to accumulate inside the polar vortex. In contrast, the negative $HNO_3$ anomaly is primarily attributed to denitrification caused by the sedimentation of NAT particles.

It is worth noting that ice PSCs can also sediment, leading to dehydration in the lower stratosphere. However, due to the high temperatures in the Arctic stratosphere, ice PSCs rarely form in the NH polar region (Tritscher et al., 2021), and thus dehydration events are rare.

        Through six sensitivity analyses, we find that the temperature anomalies between the WQBO and EQBO phases are the primary cause of PSC anomalies. During the WQBO phase, PSCs are more sensitive to temperature changes, while $H_2O$

anomalies have a smaller impact on PSCs. $HNO_3$ anomalies affect PSCs in late winter and early spring. However, some studies use fixed $H_2O$ and $HNO_3$ to diagnose PSC, an approach that ignores the effects of changes in $H_2O$ and $HNO_3$ on PSC caused by QBO, as well as other processes. It should be noted that in this study, we only considered the direct effects of $H_2O$ changes on PSCs, without considering the indirect effects of $H_2O$-induced radiative cooling of the stratosphere, which could further influence the PSC formation. As an important trace gas in the stratosphere, $H_2O$ not only affects chemical reactions but also

contributes to the radiative cooling of the stratosphere (Bi et al., 2011). Forster and Shine (2002) showed that a 1 ppmv increase in stratospheric $H_2O$ results in a 0.8 K decrease in the temperature of the tropical lower stratosphere, with a more pronounced





cooling of 1.4 K at high latitudes. Similarly, Tian et al. (2009) found that a 2 ppmv increase in $H_2O$ causes a temperature decrease of more than 4 K in the stratosphere at high latitudes. Therefore, the indirect effects of $H_2O$ on PSCs by influencing temperature may be larger than its direct effects. In our study, since the model is offline and uses meteorological fields derived

from the ERA5 reanalysis data, the changes in temperature affecting PSCs (Fig. 11) include the indirect effects of $H_2O$ on PSCs. Furthermore, the negative $HNO_3$ anomaly during the WQBO phase is due to the sedimentation of PSC particles, which not only affects the formation of PSCs in late winter and early spring but also prolongs the lifetime of active chlorine, leading to increased ozone depletion. Feng et al. (2011) found that, compared to simulations of the 2004/2005 Arctic winter without denitrification, denitrification increased ozone depletion by about 30%. This suggests that QBO-induced $HNO_3$ anomalies also

affect Arctic stratospheric ozone by affecting the lifetime of active chlorine.

The impact of the QBO on Arctic PSCs exhibits zonal asymmetry. Influenced by the QBO-induced displacement of the NH polar vortex, the centre of the negative temperature anomaly is located approximately 90° west of the climatological minimum temperature (Fig. 7). Since PSC anomalies between the WQBO and EQBO phases are primarily driven by temperature anomalies, PSCs tend to form in the region between the temperature minima and negative anomaly centre, which

is consistent with the results shown in Figs. 3 and 4. In addition, the shift of the polar vortex leads to positive temperature anomalies north of the Eurasian continent during the EQBO phase. However, because this positive anomaly centre is further away from the temperature minimum, it has little impact on PSC formation. As a result, the northern Eurasian continent has a relatively non-significant negative anomaly in PSC occurrence frequency.

This study investigates the impact of the QBO on Arctic PSC occurrence. Given the increasing frequency of QBO

disruptions and the weakening of QBO amplitude in the lower stratosphere due to climate change (Diallo et al., 2022; Wang et al., 2022), understanding how the QBO influences PSCs is crucial for predicting future changes in Arctic stratospheric chemical ozone depletion. This study provides a new perspective on ozone prediction.

**Data availability:** ERA5 data are available at https://cds.climate.copernicus.eu/datasets. The MLS data are available https://search.earthdata.nasa.gov/search. The CALIPSO dataset is available at https://asdc.larc.nasa.gov/data/CALIPSO. The ONI ca
n be obtained from https://origin.cpc.ncep.noaa.gov/products/analysis_monitoring/ensostuff/ONI_v5.php

**Author contributions:** All authors designed the study. DL and JZ performed the data analysis and prepared the paper. ZW and SZ performed the model simulations. JZ, WF and MPC contributed to the revisions made to the paper. All authors participated in the discussions and made suggestions which were considered for the final draft.

**Competing interests:** The authors declare that they have no conflicts of interest.

**Acknowledgements:** This research is supported by the National Natural Science Foundation of China (U2442211, 42130601). We gratefully acknowledge the scientific teams for CALIPSO PSC data. We would like to thank the MLS team for providing their data. We



also thank the ECMWF for providing ERA5 data. We appreciate the computing support provided by Supercomputing Center of Lanzhou University.

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
