# Peer review of "The Impact of the Stratospheric Quasi-Biennial Oscillation on Arctic Polar Stratospheric Cloud Occurrence"

_EGUsphere, 2025_

## Author Comment (AC1)

**Response to Referee's Comments**

**Manuscript No.: egusphere-2025-955**

**Title:** The Impact of the Stratospher

ic Quasi-Biennial Oscillation on Arctic Polar Stratospheric

Cloud Occurrence

**Author(s):** Douwang Li[1], Zhe Wang[1], Siyi Zhao[1], Jiankai

Zhang[1]\*, Wuhu Feng[2,3], Martyn P. Chipperfield[3,4]

**August 2025**

**Summary of revision in manuscript**

We thank very much the three reviewers for their helpful comments. We have modified our manuscript based on the comments and suggestions, which have greatly improved our paper and made it more informative. Our point-by-point replies are summarized below:

1. We acknowledge the reviewer's view that the composited PSC anomalies between the WQBO and EQBO phases may not solely result from QBO forcing. To better isolate QBO-induced PSC anomalies, we performed ensemble sensitivity experiments using the CESM model with QBO forcing. The results support the conclusion that PSC area is generally larger during the WQBO phase than during the EQBO phase.

2. As suggested, we have divided Section 3 into two subsections to improve clarity.

3. We have compared MIPAS PSC observations (2002–2012) with CALIPSO and SLIMCAT. The three datasets exhibit consistent interannual variability in PSCs, which strengthens the credibility of our conclusions.

4. Some sentences have been rephrased and the grammar has been improved.

**Response to Comments of Reviewer #1**

This manuscript investigates the impact of the stratospheric quasi-biennial oscillation (QBO) on the occurrence of Arctic polar stratospheric clouds (PSCs). This is an interesting topic, as PSCs play a critical role in ozone depletion. The QBO is a major mode of variability in the tropical stratosphere and its effects on the polar vortex and ozone have been studied, but its specific impact on PSCs has not been explored in depth. Therefore, this study fills a gap in the existing literature and is innovative. In this study, using the CALIPSO satellite observations and the SLIMCAT chemical transport model, the authors found that QBO can have a significant effect on the Arctic PSC, characterized by a clear zonal asymmetry. Moreover, the authors also found that QBO affects Arctic $H_2O$ and $HNO_3$ in two different ways. These conclusions are based on observations and simulations and appear reasonable. Overall, this paper is well written. However, part of the analysis needs to be clarified and improved. I encourage the authors to revise it before publication.

**General comments:**

1. You mentioned that during the EQBO phase, the distribution of the PSC area is skewed, with a peak near zero. This is understandable, as the polar vortex is generally weaker and the temperatures inside the vortex are higher during the EQBO phase, which is unfavorable for PSC formation. However, during the WQBO phase, the polar vortex is stronger and the temperatures inside the vortex are lower, which favors PSC formation. So why is the distribution of the PSC area during the WQBO phase not a skewed distribution with a higher peak, but rather a uniform distribution?

**Response: Thank you for your comment. As you mentioned, the polar vortex is generally stronger and colder during the WQBO phase, and weaker and warmer during the EQBO phase. Since the QBO influences Arctic PSCs primarily through its modulation of stratospheric temperature, PSC area is typically smaller during**

EQBO and larger during WQBO, which is consistent with our findings.

To understand the distribution patterns, we examined the probability density functions (PDF) of Arctic stratospheric temperature and polar vortex strength (defined by the zonal-mean zonal wind at 500 K isentropic level) during the WQBO, EQBO, and neutral QBO (NQBO) phases (Figure R1). The results show that extremely low temperatures and strong vortex events are rare during all QBO phases, suggesting that the frequency of extreme PSC occurrences will be limited.

We further analyzed the PDF of PSC volume during the WQBO and EQBO phases (see Figure 1 in the manuscript), and find that both distributions exhibit a skewed pattern and the frequency of extreme PSC is very small, which is consistent with theoretical expectations. Although the shape of the distribution may differ depending on the metric used (area vs. volume), the key conclusion remains robust: PSC occurrence is more frequent and more extensive during the WQBO phase compared to the EQBO phase.

[Figure]

**Figure R1.** The probability distribution functions (PDF) of (a) temperature over the Arctic region (60°N–90°N) and (b) zonal-mean zonal wind between 60°N and 65°N during different QBO phases at 500 K isentropic level. Red lines represent the WQBO phase, blue lines represent the EQBO phase, and black lines represent the NQBO phase.

2. I suggest dividing Section 3 into several subsections to improve the structural clarity and logical coherence of the manuscript, such as (1) The impact of QBO on PSCs; and (2) The key factors responsible for QBO's impact.

**Response: Thank you for your suggestion. We agree that dividing Section 3 into several subsections would improve the structural clarity and logical coherence of the manuscript. In the revised version, we divide Section 3 into two subsections: (1) Impact of QBO on Arctic PSCs; and (2) How the QBO influences the Arctic PSC area.**

3. The current analysis relies on a single observational dataset (CALIPSO), which has a relatively limited temporal coverage. Are there other PSC observations covering different time periods that could be incorporated? I suggest the authors consider using additional observational datasets to further validate the key conclusion of the manuscript, that PSC occurrence is more frequent during WQBO phases compared to EQBO phases. This would help enhance the robustness and credibility of the findings.

**Response: Thank you for your insightful suggestion. The Michelson Interferometer for Passive Atmospheric Sounding (MIPAS) observed PSCs over both polar regions from 2002 to 2012 (Spang et al., 2018). To complement our CALIPSO-based analysis, we apply the "P18 method" to MIPAS data. We then perform composite analyses of the PSC area between the WQBO and EQBO phases. As shown in Figure R2, the results indicate a larger PSC area during the WQBO phase compared to the EQBO phase. We note that there are fewer points that pass the significance test, which may be due to limited sample size (5 samples for the WQBO and 3 samples for the EQBO). For this reason, we do not include the MIPAS composite results in the main manuscript. We only used the MIPAS data to validate the robustness of the CALIPSO and SLIMCAT PSC data, thereby enhancing the credibility of our results.**

[Figure]

**Figure R2. Differences in Arctic PSC area between the WQBO and EQBO phases derived from MIPAS during December–March (DJFM) and the DJFM average. Solid filled symbols indicate the differences are statistically significant at the 95% confidence level according to the Student's _t_-test.**

**To further support our conclusions, we calculated the PSC volume by vertically integrating the PSC coverage area from CALIPSO, MIPAS, and SLIMCAT. Figure R3 shows the time series of Arctic PSC volume from these three datasets. Due to CALIPSO's higher detection threshold, the observed PSC volume is smaller than those of MIPAS and SLIMCAT. Nevertheless, the interannual variability is remarkably consistent across all three datasets. Table 1 summarizes the PSC volumes during the WQBO and EQBO phases, showing that all three datasets indicate significantly greater PSC volume during the WQBO phase compared to the EQBO phase. Overall, the agreement between satellite observations (CALIPSO and MIPAS) and SLIMCAT simulations provides strong support for the robustness of our key conclusion.**

[Figure]

**Figure R3. (a)** Interannual variation of Arctic PSC volume (December–March mean) anomalies and **(b)** daily evolution of Arctic PSC volume observed by CALIPSO, MIPAS, and simulated by SLIMCAT. In the horizontal axis, blue and red labels indicate EQBO and WQBO winters, respectively. In panel (a), the different colours on the vertical axis represent different data sources.

**Table 1. December–March mean PSC volumes for CALIPSO, MIPAS, and SLIMCAT during WQBO and EQBO phases and differences between the WQBO and EQBO phases (unit: $10^6$ km$^3$). The volumes are calculated over the vertical range of 10–30 km.**

|  | CALIPSO | MIPAS | SLIMCAT |
|---|---|---|---|
| WQBO | 10.82 | 19.13 | 42.29 |
| EQBO | 6.68 | 10.05 | 20.08 |
| Diff | 4.14 | 9.08 | 22.21 |

**Specific comments:**

P1, L12: analyzes -> examines

**Response: Thank you for your comment. Corrected. (Please see P1 and L12 in the revised manuscript)**

P1, L13: there is -> there exists

**Response: Thank you for your comment. We changed "Previous studies have shown that there is a linear relationship between ozone loss and PSC volume illuminated by sunlight". In the sentence "there is", we have replaced "there is" with "there exists". (Please see P1 and L29 in the revised manuscript)**

P1, L27: I would suggest an explanation of Clx.

**Response: Thank you for your comment. $Cl_x$ represents reactive chlorine species, including Cl, ClO, $Cl_2O_2$ which can participate in ozone depletion. We have added a brief explanation of $Cl_x$ in the revised manuscript: (Please see P1 and L27-L28 in the revised manuscript)**

*"When spring arrives, these reactive chlorine atoms participate in the $Cl_x$ (= Cl + ClO + $2Cl_2O_2$) catalytic cycles that destroy stratospheric ozone (Solomon et al., 1986, 2015)."*

P1, L29: Delete the "." after sunlight.

**Response: Thank you for your comment. Deleted.**

P2, L47: Add a comma after "HNO3".

**Response: Thank you for your comment. Added. (Please see P2 and L48 in the revised manuscript)**

P2, L49: atmospheric-> atmosphere

**Response: Thank you for your comment. Corrected. (Please see P2 and L50 in the revised manuscript)**

P3, L69-L70: SSW->SSWs; which have -> which has

**Response: Thank you for your comment. Corrected. (Please see P3 and L70-L71 in the revised manuscript)**

P4, L22: spans -> span

**Response: Thank you for your comment. Corrected. (Please see P4 and L127 in the revised manuscript)**

P5, L133: vertical range spanning from 316 to 0.00215 hPa -> vertical range of 316 hPa to 0.00215 hPa; Delete "an".

**Response: Thank you for your comment. Corrected. (Please see P5 and L138 in the revised manuscript)**

P6, L163: surface density-> surface area density

**Response: Thank you for your comment. Corrected. (Please see P6 and L168 in the revised manuscript)**

P6, L187: Are the PSC coverage areas of SLIMCAT and CALIPSO daily? You need clarify.

**Response: Thank you for your comment. The PSC coverage areas of SLIMCAT and CALIPSO are daily. We have clarified it: (Please see P7 and L193 in the revised manuscript)**

*"In this study, two different methods are used to calculate the daily PSC coverage areas of CALIPSO and SLIMCAT, respectively."*

P8, L215: on 500 K -> on the 500 K

**Response: Thank you for your comment. We have modified Figure 1 and deleted "on 500 K".**

P8, L221: How do you perform the composite analyses, was it WQBO-EQBO?

**Response: Thank you for your question. We calculated the difference between the WQBO and EQBO phases (WQBO minus EQBO). We clarify it in the revised manuscript: (Please see P9 and L240 in the revised manuscript)**

*"To investigate the relationship between the QBO and PSCs, composite analyses of*

*PSCs are performed (WQBO minus EQBO)."*

P8; L221-222: are removed -> were removed

**Response: Thank you for your comment. Corrected. (Please see P9 and L242 in the revised manuscript)**

P8, L226: level -> levels

**Response: Thank you for your comment. Corrected. (Please see P9 and L245 in the revised manuscript)**

P8, L227: differences -> differences in PSC area

**Response: Thank you for your comment. Corrected. (Please see P9 and L246 in the revised manuscript)**

P8, L228: Why are the differences in the SLIMCAT PSC area larger than those observed by CALIPSO? Does SLIMCAT reproduce the observed PSCs well?

**Response: Thank you for your insightful question. The differences in PSC area between the WQBO and EQBO phases are larger in the SLIMCAT simulations than in the CALIPSO observations. This may be since the PSC area of the SLIMCAT is much larger than that of the CALIPSO. There are several factors contributing to the discrepancy between simulated and observed PSCs (Li et al., 2024). First, although CALIPSO has high vertical and horizontal resolution, its limitations in detecting optically thin clouds may lead to an underestimation of the PSC due to the high detection threshold. Second, the lower spatial resolution of SLIMCAT leads to an overestimation of the PSC coverage area.**

**Although the PSC area and occurrence frequency simulated by SLIMCAT are larger than those observed by CALIPSO, SLIMCAT captures the key features of PSC variability well, including the seasonal cycle, interannual variability, and spatial patterns (Li et al., 2024). In addition, SLIMCAT reproduces well the**

enhanced PSC area and occurrence frequency during the WQBO phase compared to the EQBO phase, consistent with CALIPSO observations.

We added this sentence to the manuscript: **(Please see P9 and L249-L251 in the revised manuscript)**

*"The greater differences in SLIMCAT PSC area between the WQBO and EQBO phases primarily result from SLIMCAT simulating larger PSC areas than CALIPSO observations, likely due to CALIPSO's higher detection threshold (Li et al., 2024)."*

References:

Li, D., Wang, Z., Li, S., Zhang, J., and Feng, W.: Climatology of Polar Stratospheric Clouds Derived from CALIPSO and SLIMCAT, Remote Sens., 16, 3285, https://doi.org/10.3390/rs16173285, 2024.

P10, L254: zonal asymmetry of -> zonal asymmetry in

**Response: Thank you for your comment. Corrected. (Please see P11 and L275 in the revised manuscript)**

P10, L255: changes -> shifts

**Response: Thank you for your comment. Corrected. (Please see P11 and L276 in the revised manuscript)**

P10, L266: that in -> those in

**Response: Thank you for your comment. Corrected. (Please see P11 and L287 in the revised manuscript)**

P12, L291: during 1979–2022 -> for the period 1979–2022

**Response: Thank you for your comment. Corrected. (Please see P13 and L312 in the revised manuscript)**

P13, L314: Arctic -> the Arctic

**Response: Thank you for your comment. Corrected. (Please see P14 and L336 in the revised manuscript)**

P18, L409: strength-> the strength

**Response: Thank you for your comment. Corrected. (Please see P19 and L433 in the revised manuscript)**

---

## Author Comment (AC2)

**Response to Referee's Comments**

**Manuscript No.: egusphere-2025-955**

**Title:** The Impact of the Stratospheric Quasi-Biennial

Oscillation on Arctic Polar Stratospheric Cloud Occurrence

**Author(s):** Douwang Li[1], Zhe Wang[1], Siyi Zhao[1], Jiankai

Zhang[1]\*, Wuhu Feng[2,3], Martyn P. Chipperfield[3,4]

**August 2025**

**Summary of revision in manuscript**

We thank very much the three reviewers for their helpful comments. We have modified our manuscript based on the comments and suggestions, which have greatly improved our paper and made it more informative. Our point-by-point replies are summarized below:

1. We acknowledge the reviewer's view that the composited PSC anomalies between the WQBO and EQBO phases may not solely result from QBO forcing. To better isolate QBO-induced PSC anomalies, we performed ensemble sensitivity experiments using the CESM model with QBO forcing. The results support the conclusion that PSC area is generally larger during the WQBO phase than during the EQBO phase.

2. As suggested, we have divided Section 3 into two subsections to improve clarity.

3. We have compared MIPAS PSC observations (2002–2012) with CALIPSO and SLIMCAT. The three datasets exhibit consistent interannual variability in PSCs, which strengthens the credibility of our conclusions.

4. Some sentences have been rephrased and the grammar has been improved.

**Response to Comments of Reviewer #2**

General comments:

This manuscript, "The impact of the Stratospheric Quasi-Biennial Oscillation on Arctic Polar Stratospheric Occurrence", by Li et al. investigates how the QBO modulates Arctic PSC formation and, by extension, ozone depletion processes. The paper addresses a relatively underexplored but important linkage between QBO phase and PSC variability using satellite observations and SLMICAT simulations, and separates the roles of temperature, $H_2O$ and $HNO_3$, via sensitivity experiments. It is topical given the implications for Arctic ozone recovery.

Specific comments:

1. My major concern lies with the sensitivity experiments. While the use of composites to assess QBO-related anomalies is methodologically reasonable, it remains unclear whether the observed changes in temperature (T), water vapor ($H_2O$), and nitric acid ($HNO_3$) are causally induced by the QBO alone.

In reality, the anomalies during WQBO/EQBO phases are influenced by multiple factors, and not solely by the QBO. Thus, attributing the entire composite difference to QBO forcing may overestimate its impact. I recommend that the authors address how they isolate QBO-specific variability from other confounding influences (e.g., ENSO, solar variability, volcanic aerosols).

If a full isolation is not feasible with the current dataset, the authors should clearly acknowledge this limitation and discuss the potential implications on the interpretation of their sensitivity results.

**Response: Thank you for your constructive comment. We agree that other factors such as ENSO, solar variability, and volcanic aerosols may also influence the observed changes in temperature (T), water vapour ($H_2O$), and nitric acid ($HNO_3$).**

To address this concern, we conducted two groups of ensemble experiments using the CESM1.2.2 model with interactive stratospheric chemistry.

We generated two 40-member ensembles: WQBO run and EQBO run. We first compute the composite mean meteorological fields during the WQBO and EQBO phases using MERRA2 reanalysis data. Stratospheric zonal wind, meridional wind, and temperature between 22°S–22°N and 84–4 hPa (Hansen et al., 2013) are nudged to these composite mean fields. Initial conditions for the 40 ensemble members were obtained from a 40-year free-running simulation of the CESM FW component, using outputs from December 1 of each year. Both ensemble experiments were integrated from December 1 to March 31. Importantly, both ensembles used the same sea surface temperatures, solar flux, and volcanic aerosol forcing, thereby minimizing the influences of ENSO, solar variability, and volcanic eruptions.

Figure R1 shows the difference in PSC area between the two ensemble runs. The PSC area during the WQBO phase is significantly larger than that during the EQBO phase, with the largest difference occurring near the 500 K isentropic level, consistent with both CALIPSO observations and SLIMCAT simulations (see Figure 2 in the manuscript). While the SLIMCAT results show a large PSC anomaly starting in December, CESM shows the anomaly emerging in January, likely because nudging began on December 1 and the polar vortex requires time to respond to QBO forcing. We analyzed the differences in temperature between the WQBO run and EQBO run. As shown in Figure R2, CESM reproduces the cold anomaly during WQBO, although the magnitude is smaller than that in ERA5. This temperature underestimation likely explains the weaker PSC area response (Figure R1 and Figure 2c in the manuscript).

[Figure]

**Figure R1. Differences in PSC coverage area between the WQBO run and the EQBO run. Black dotted regions indicate the differences in PSC area are statistically significant at the 95% confidence level according to the Student's *t*-test.**

[Figure]

**Figure R2. Time evolution of polar temperature differences between the WQBO and EQBO phases from (a) CESM and (b) ERA5. Black dotted regions indicate the differences in temperature are statistically significant at the 95% confidence level according to the Student's *t*-test.**

**We further compare the differences in H₂O and HNO₃ from CESM simulation (Figure R3), MLS observations (Figure R4), and SLIMCAT simulations (Figure R5) between the WQBO and EQBO phases. Both MLS and SLIMCAT show a positive H₂O anomaly between the 500–700 K isentropic levels from December to March. In contrast, the positive H₂O anomaly in CESM persists only through January. For HNO₃, both MLS and SLIMCAT show a negative anomaly between 400 and 500 K during WQBO, which is caused by the denitrification process. However, the CESM shows positive anomaly in HNO₃. This is likely due to the incomplete representation of denitrification and dehydration processes in CESM.**

In addition to the simplified representation of denitrification and dehydration, we acknowledge that internal feedback mechanisms in CESM may also contribute to the HNO₃ anomalies. Further investigation would be needed to disentangle these effects.

[Figure]

**Figure R3. Differences in (a) H₂O and (b) HNO₃ between the WQBO run and EQBO run from CESM. The white contours are their climatological concentration. Black dotted regions indicate the differences are statistically significant at the 95% confidence level according to the Student's *t*-test.**

[Figure]

**Figure R4. Differences in (a) H₂O and (b) HNO₃ between the WQBO and EQBO phases from MLS observations. The white contours are their climatological concentration. Black dotted regions indicate the differences are statistically significant at the 95% confidence level according to the Student's _t_-test.**

[Figure]

**Figure R5. Differences in (a) H₂O and (b) HNO₃ between the WQBO and EQBO phases from SLIMCAT observations. The white contours are their climatological concentration. Black dotted regions indicate the differences are statistically significant at the 95% confidence level according to the Student's *t*-test.**

**These comparisons reveal that while CESM can qualitatively reproduce the temperature response to QBO, it underestimates the impact on PSCs and fails to capture the observed H₂O and HNO₃ responses. Therefore, using CESM for further sensitivity experiments would introduce greater uncertainties. For this reason, we did not conduct further sensitivity experiments using the CESM. We acknowledge that the composite differences derived from reanalysis and SLIMCAT simulations may still include influences from other processes. We have added a clear statement of this limitation and its implications for interpretation in the revised manuscript. We added this paragraph to the revised manuscript: (Please see P23 and L524-L530 in the revised manuscript)**

*"Although our study provides evidence for the influence of QBO phases on Arctic PSC occurrence and associated stratospheric chemical composition changes, several limitations must be acknowledged. First, the SLIMCAT model used in this study is an offline chemical transport model driven by reanalysis data. Despite prescribing fixed ODS as well as climatological SAD and solar flux in the simulations, it is impossible to fully isolate the QBO signal from other factors such as ENSO, solar variability, and volcanic aerosols. Although composite analyses help to reveal QBO-related effects, the resulting anomalies may still contain contributions from other climate factors, which may contaminate the impact of QBO on Arctic PSC occurrence."*

**References:**

**Hansen, F., Matthes, K., and Gray, L. J.: Sensitivity of stratospheric dynamics and chemistry to QBO nudging width in the chemistry–climate model WACCM, J. Geophys. Res.: Atmos., 118, 10,464-10,474, https://doi.org/10.1002/jgrd.50812, 2013.**

2. ENSO is a major interannual variability of the NH polar vortex. The authors show that the significance of the PSC occurrence anomalies during W/E-QBO is substantially reduced after excluding strong ENSO years (Fig. 5). However, it remains unclear how much of the observed differences are independently attributable to the QBO versus ENSO. Would the results change after regressing out the ENSO variability? I recommend that the authors clarify the influence of ENSO years quantitatively.

**Response: Thank you for your valuable suggestion. As recommended, we applied a linear regression approach to remove the influence of ENSO. Specifically, we regressed the PSC occurrence frequency against the ONI index and obtained the regression coefficient a and intercept b, which were then used to calculate the residuals:**

$$PSC_{residual} = PSC - (a \times ONI + b)$$

Here, *PSC* represents the original occurrence frequency, and $PSC_{residual}$ denotes the PSC occurrence frequency without the ENSO signal.

Figure R6 shows the differences in PSC occurrence between the WQBO and EQBO phases after removing the ENSO signal. Compared to Figure 4 in the manuscript, the significant positive anomalies remain in the polar regions, and our key conclusion still holds.

[Figure]

Figure R6. Differences in PSC occurrence frequency between the WQBO and EQBO phases after removing the ENSO signal on the 500 K isentropic level derived from the SLIMCAT simulations for the period 1979–2022 for (a) winter average (December to February), (b) December, (c) January, and (d) February. Black dotted regions indicate the differences in PSC occurrence frequency are statistically significant at the 95% confidence level according to the Student's *t*-test.

To further clarify the contribution of ENSO, Figure R7 presents the ENSO-

induced component ( $a \times ONI + b$ ) in the composite difference. The results indicate that the ENSO-related PSC occurrence frequency anomalies are minimal and non-significant, with the differences between the WQBO and EQBO phases remaining within ±2%, suggesting a minor influence of ENSO.

[Figure]

**Figure R7. Differences in PSC occurrence frequency between the WQBO and EQBO phases induced by the ENSO signal on the 500 K isentropic level derived from the SLIMCAT simulations for the period 1979–2022 for (a) winter average (December to February), (b) December, (c) January, and (d) February. Black dotted regions indicate the differences in PSC occurrence frequency are statistically significant at the 95% confidence level according to the Student's *t*-test.**

To verify the robustness of our method, we also applied the same regression approach to remove the QBO signal (by regressing PSC occurrence frequency against the QBO index). The differences in PSC occurrence after removing the QBO signal are shown in **Figure R8**. The results clearly show a substantial reduction in the magnitude of significant anomalies, confirming the effectiveness and reasonableness of our regression-based method for isolating ENSO influences.

[Figure]

**Figure R8. Differences in PSC occurrence frequency between the WQBO and EQBO phases after removing the QBO signal on the 500 K isentropic level derived from the SLIMCAT simulations for the period 1979–2022 for (a) winter average (December to February), (b) December, (c) January, and (d) February. Black dotted regions indicate the differences in PSC occurrence frequency are statistically significant at the 95% confidence level according to the Student's *t*-test.**

3. It is clear that the chemical transport model (CTM) used in this study does not include interactive radiative-dynamical feedbacks.

This is an important limitation, as feedbacks between radiation, temperature, and circulation could alter the stratospheric response to QBO forcing. I recommend the authors explicitly discuss how the absence of radiative-dynamical coupling may affect their results, particularly the sensitivity experiments and the interpretation of temperature-driven PSC variability.

**Response: Thank you for your insightful comment. As you noted, SLIMCAT used in our study is an offline chemical transport model that does not incorporate the**

chemical-radiative-dynamical coupling process. However, SLIMCAT is driven by ERA5 reanalysis data, which incorporate the effects of radiative-dynamical processes. Moreover, SLIMCAT is designed to simulate realistic stratospheric chemistry as accurately as possible. Therefore, the results of the composite analysis reflect the impact of the QBO on Arctic stratospheric temperature, $H_2O$ and $HNO_3$, and the PSC anomalies that we obtained are also largely caused by the QBO.

In our sensitivity analyses, we only considered the direct impact of QBO-induced $H_2O$ changes on Arctic PSCs area, without accounting for the indirect effect by radiative cooling. In reality, as an important trace gas, stratospheric $H_2O$ has a non-negligible influence on stratospheric temperature. Considering only the direct effect of $H_2O$ changes may underestimate the overall influence of $H_2O$ on PSC area in the sensitivity analysis. We have added a discussion of this limitation and its impact on the results of the sensitivity analyses in the revised manuscript: (Please see P23 and L530-L538)

"Second, SLIMCAT does not include the chemical-radiative-dynamical coupling process. As an important trace gas in the stratosphere, $H_2O$ not only affects chemical reactions but also contributes to the radiative cooling of the stratosphere (Bi et al., 2011). Forster and Shine (2002) showed that a 1 ppmv increase in stratospheric $H_2O$ results in a 0.8 K decrease in the temperature of the tropical lower stratosphere, with a more pronounced cooling of 1.4 K at high latitudes. Similarly, Tian et al. (2009) found that a 2 ppmv increase in $H_2O$ causes a temperature decrease of more than 4 K in the stratosphere at high latitudes. In particular, due to the high sensitivity of PSC formation to temperature, the indirect effects of $H_2O$ on PSCs by influencing temperature may be comparable to its direct effects. In our sensitivity analyses, we only consider the direct effect of $H_2O$ changes on PSCs, without accounting for the indirect impact of radiative cooling induced by $H_2O$ anomalies. This omission may lead to an underestimation of the QBO's impact on the Arctic PSC area in Fig. 11e–

*h."*

Technical corrections:

1. In Figures 3 and 4 (PSC occurrence anomalies), the color scales could be optimized to better show the large positive anomalies. Currently, they are all just dark red.

**Response: Thank you for your comment. We have adjusted the color scales in Figures 3 and 4 to enhance the visibility of large positive anomalies, ensuring a clearer distinction across the full range of values. (Please see P11 L279 and P12 L299 in the revised manuscript)**

2. Abstract, Line 13: change "no studies have deeply analyzed" to "few studies have thoroughly analyzed" for a better scientific tone.

**Response: Thank you for your comment. We have revised the sentence to "few studies have thoroughly analyzed" to improve the scientific tone. (Please see P1 and L11-L12 in the revised manuscript)**

3. Line 157: "shown accurately simulate" missing "to".

**Response: Thank you for your comment. We have added "to". (Please see P6 and L162 in the revised manuscript)**

4. Line 205: Repeating "samples"

**Response: Thank you for your comment. The sentences are rephrased in the revised paper: (Please see P8 and L220-L221 in the revised manuscript)**

*"For example, of the 14 SLIMCAT simulations, 5 show values below 10 million km$^3$. In contrast, during the WQBO phase, all SLIMCAT simulations exceed 10 million km$^3$."*

5. Line 240: "variation overtime on the left panel". Typo: should be "over time"

**Response: Thank you for your comment. Corrected. (Please see P10 and L261 in**

**the revised manuscript)**

6. Line 409: "as strength of the polar vortex increases" --> "as the strength of the polar vortex increases"

**Response: Thank you for your comment. Corrected. (Please see P19 and L433 in the revised manuscript)**

7. Line 320: "Figure. 7" --> "Figure 7".

**Response: Thank you for your comment. Corrected. (Please see P15 and L342 in the revised manuscript)**

---

## Author Comment (AC3)

**Response to Referee's Comments**

**Manuscript No.: egusphere-2025-955**

**Title:** The Impact of the Stratospheric Quasi-Biennial

Oscillation on Arctic Polar Stratospheric Cloud Occurrence

**Author(s):** Douwang Li[1], Zhe Wang[1], Siyi Zhao[1], Jiankai

Zhang[1]*, Wuhu Feng[2,3], Martyn P. Chipperfield[3,4]

**August 2025**

**Summary of revision in manuscript**

We thank very much the three reviewers for their helpful comments. We have modified our manuscript based on the comments and suggestions, which have greatly improved our paper and made it more informative. Our point-by-point replies are summarized below:

1. We acknowledge the reviewer's view that the composited PSC anomalies between the WQBO and EQBO phases may not solely result from QBO forcing. To better isolate QBO-induced PSC anomalies, we performed ensemble sensitivity experiments using the CESM model with QBO forcing. The results support the conclusion that PSC area is generally larger during the WQBO phase than during the EQBO phase.

2. As suggested, we have divided Section 3 into two subsections to improve clarity.

3. We have compared MIPAS PSC observations (2002–2012) with CALIPSO and SLIMCAT. The three datasets exhibit consistent interannual variability in PSCs, which strengthens the credibility of our conclusions.

4. Some sentences have been rephrased and the grammar has been improved.

**Response to Comments of Reviewer #3**

Peer review

The manuscript investigates the influence of the quasi-biennial oscillation (QBO) on the occurrence of Arctic polar stratospheric clouds (PSCs) using CALIPSO satellite observations (2006-2021) and SLIMCAT model simulations (1979-2022). The study shows that PSC coverage is significantly larger during the westerly QBO (WQBO) phase compared to the easterly QBO (EQBO) phase, with a zonally asymmetric anomaly pattern. The authors analyze the mechanisms driving these differences, attributing them primarily to QBO-induced temperature changes, with secondary contributions from water vapor and nitric acid variations. Sensitivity tests further emphasize the dominant role of temperature.

The topic is highly relevant for understanding polar stratospheric chemistry and ozone depletion processes under future climate scenarios. The combined use of long-term satellite observations and chemical transport modeling is a strong methodological approach. The manuscript is generally well-structured, clearly written, and supported by comprehensive references. The sensitivity analysis provides valuable insight into the relative contributions of temperature, $H_2O$, and $HNO_3$.

I really enjoyed reading this work, and I believe it definitely deserves to be published. The manuscript is scientifically sound, well-presented, and makes a valuable contribution to the understanding of stratospheric processes. However, he authors might consider expanding it along the lines of the suggestions listed below.

General comments:

The CALIPSO dataset (16 years) is relatively short for robust statistical analysis, as noted by the authors. While SLIMCAT compensates with a longer timeframe, the observational validation remains limited. The authors may discuss potential biases or uncertainties arising from the short observational record and how SLIMCAT's longer

simulations mitigate this. Moreover, statistically significant differences related to the QBO phase appear over regions hosting important ground-based lidar stations with long-term data records. Have the authors tried to verify their findings by also making use of these datasets and/or referring to published results?

**Response: We appreciate the reviewer's insightful comment. The CALIPSO PSC dataset spans only 15 Arctic winters, which limits the robustness of the composite analyses. The results may be influenced by extreme events in specific years. To address this limitation, our study utilized the SLIMCAT long-term simulation (1979–2022) to supplement the relatively short observational record. These simulations help reduce the influence of individual outlier events on the composite analysis and enhance the statistical significance of the results. We have added a clarification of this issue in the revised manuscript. (Please see P4 L103-L109 and P22 L480-L488)**

*"From 2006 to 2021, the Cloud-Aerosol Lidar and Infrared Pathfinder Satellite Observations (CALIPSO) mission continuously observed PSCs over both the Arctic and Antarctic, providing an unprecedented view of PSC occurrence and composition (Tritscher et al., 2021). In this study, we utilize CALIPSO PSC observations (Pitts et al., 2018) to investigate the potential impact of the QBO on Arctic PSC occurrence. However, the CALIPSO record includes only 15 Arctic winters, which may limit the statistical robustness of the results—for instance, the results could be affected by extreme events. To address this limitation, we also incorporate simulations from the SLIMCAT 3D chemical transport model, which spans over 40 years from 1979 to 2022, to complement the observational analysis."*

*"It is important to note that when the sample size is small, composite analysis results may be influenced by individual extreme events. For example, the negative anomaly in the PSC area derived from CALIPSO in December may have been driven by a few specific years, which is contrary to the theoretical expectations. To address this issue,*

*we used SLIMCAT simulations for the period 1979–2022 to reduce the impact of individual extreme events on the composite results. The results show that with the extension of the simulation period, a positive anomaly consistent with theoretical expectations occurs in December, and the statistical significance of the composite analysis is improved."*

Furthermore, we acknowledge the value of long-term ground-based lidar records at key Arctic locations. As noted by Tesche et al. (2021), among Arctic sites suitable for PSC observations and with published PSC data, only Eureka and Ny-Ålesund have reported long-term lidar-based PSC measurements. However, we were unable to obtain the datasets from these sites. For instance, while Tritscher et al. (2021) included Ny-Ålesund lidar observations spanning 1995–2018 (see their Fig. 25), the underlying data were not accessible to us. We also contacted researchers at the Alfred Wegener Institute to request the Ny-Ålesund dataset, but unfortunately, we did not obtain the data.

Given these limitations, we try to verify our findings through comparison with published results and the Michelson Interferometer for Passive Atmospheric Sounding (MIPAS) observations. To complement our CALIPSO-based analysis, we apply the "P18 method" to MIPAS data. We then perform composite analyses of the PSC area between the WQBO and EQBO phases. As shown in Figure R1, the results indicate a larger PSC area during the WQBO phase compared to the EQBO phase. We note that there are fewer points that pass the significance test, which may be due to limited sample size (5 samples for the WQBO and 3 samples for the EQBO). For this reason, we do not include the MIPAS composite results in the main manuscript. We only used the MIPAS data to validate the robustness of the CALIPSO and SLIMCAT PSC data, thereby enhancing the credibility of our results.

[Figure]

**Figure R1. Differences in Arctic PSC area between the WQBO and EQBO phases derived from MIPAS during December–March (DJFM) and the DJFM average. Solid filled symbols indicate the differences are statistically significant at the 95% confidence level according to the Student's *t*-test.**

To further support our conclusions, we calculated the PSC volume by vertically integrating the PSC coverage area from CALIPSO, MIPAS, and SLIMCAT. Figure R2 presents the time series of PSC volume from CALIPSO and MIPAS satellite observations, alongside SLIMCAT simulations. Due to the higher detection threshold, CALIPSO's PSC volume is systematically lower than that of MIPAS and SLIMCAT. Nevertheless, the interannual variability in PSC volume is remarkably consistent across all three datasets. Moreover, the interannual variation in SLIMCAT PSC volume is generally consistent with the interannual variation in the PSC sighting frequencies at Ny-Ålesund as reported in Tritscher et al. (2021, Fig. 25). Inconsistent changes in some years may be due to the Ny-Ålesund site data can only represent PSC changes at the site location and do not

**represent PSC changes in the entire Arctic region.**

[Figure]

**Figure R2. (a) Interannual variation of Arctic PSC volume (December–March mean) anomalies and (b) daily evolution of Arctic PSC volume observed by CALIPSO, MIPAS, and simulated by SLIMCAT. In the horizontal axis, blue and red labels indicate EQBO and WQBO winters, respectively. In panel (a), the different colours on the vertical axis represent different data sources.**

**References:**

Tesche, M., Achtert, P., and Pitts, M. C.: On the best locations for ground-based polar stratospheric cloud (PSC) observations, Atmos. Chem. Phys., 21, 505–516, https://doi.org/10.5194/acp-21-505-2021, 2021.

Tritscher, I., Pitts, M. C., Poole, L. R., Alexander, S. P., Cairo, F., Chipperfield, M. P., Grooß, J., Höpfner, M., Lambert, A., Luo, B., Molleker, S., Orr, A., Salawitch, R., Snels, M., Spang, R., Woiwode, W., and Peter, T.: Polar Stratospheric Clouds: Satellite Observations, Processes, and Role in Ozone Depletion, Rev. Geophys., 59, https://doi.org/10.1029/2020RG000702, 2021.

The SLIMCAT model uses simplified PSC schemes (e.g., fixed number densities for NAT/ice particles). How might this affect the representation of denitrification/dehydration processes? The authors may speculate on how more sophisticated microphysics (e.g., size-resolved NAT sedimentation) would alter the

conclusions.

**Response: Thank you for your insightful comment regarding the simplified PSC scheme in SLIMCAT. We have added a discussion in the revised manuscript acknowledging the potential limitations of using prescribed particle radius or number densities for NAT and ice particles. While this simplification may affect the representation of denitrification and dehydration, we argue that the key QBO–PSC relationships reported in our study remain robust, as they are primarily driven by temperature. Nevertheless, we agree that implementing a more sophisticated microphysical scheme (e.g., including NAT/ice particle growth and sedimentation) would be a valuable extension in future work. The following statement was added to the discussion: (Please see P23 and L538-L544 in the revised manuscript)**

*"Finally, in SLIMCAT, denitrification and dehydration are implemented by assuming fixed sedimentation velocities for NAT and ice particles based on prescribed particle radii or number densities. This simplified scheme still shows discrepancies in $H_2O$ and $HNO_3$ compared to MLS observations. Incorporating more complex microphysical schemes, such as the DLAPSE, which incorporates the nucleation, growth, and settlement processes of PSC particles, could improve the simulation of the spatial distribution of $H_2O$ and $HNO_3$. However, detailed microphysical schemes are too expensive for long-term simulations. Moreover, as PSC formation is primarily modulated by temperature, the relationship between QBO and PSCs established in this study remains robust."*

The authors dismiss BD circulation as the driver of $H_2O$ anomalies but do not fully explore alternative mechanisms. As instance, the may clarify whether the $H_2O$ accumulation is purely due to vortex isolation or if other processes (e.g., local tropopause temperature and permeability changes) may contribute.

**Response:** Thank you for your comment. We have examined the QBO-related differences in MLS $H_2O$ between EQBO and WQBO phases during December–March over the 300–1 hPa (Figure R3). Our study focuses on the region north of 60 °N and around 30 hPa, where we observe statistically significant positive $H_2O$ anomalies. In the manuscript, we attribute this anomaly primarily to vortex isolation. You mentioned the possibility that local tropopause temperature and permeability changes could also contribute. In polar regions, the tropopause is typically located near 300–200 hPa. We do observe that QBO significantly affects $H_2O$ concentrations near the tropopause. However, in the region between the tropopause and our study level (30 hPa), the response of $H_2O$ to the QBO exhibits inconsistent signs, indicating that the vertical variation in the $H_2O$ is not continuous. Therefore, it is unlikely that the $H_2O$ anomalies observed at 30 hPa are primarily influenced by local changes near the tropopause. The following sentences were added in the revised paper: (Please see P16 and L387-L391 in the revised manuscript)

*"Changes in tropopause temperature and permeability may also influence stratospheric $H_2O$. In the Arctic, the tropopause is typically located around the 320 K potential temperature level. However, we note that the positive $H_2O$ anomalies observed and simulated in our study are mainly concentrated above the 450 K, with no significant positive $H_2O$ anomaly signals detected on 320 K–450 K (Fig. 8a and c). Therefore, we consider that local processes at the tropopause are unlikely to be the primary drivers of the $H_2O$ anomalies above the 450 K."*

[Figure]

**Figure R3.** Climatological $H_2O$ (white contours) and the differences in zonal $H_2O$ between WQBO and EQBO phases (shadings, WQBO–EQBO) derived from MLS data for the period 2004−2021. Black dotted regions indicate the differences in $H_2O$ are statistically significant at the 95% confidence level according to the Student's *t*-test.

In addition, we have considered two other potential contributions to the $H_2O$ anomalies in the manuscript:

(1) Methane oxidation, which contributes to $H_2O$ in the middle and upper stratosphere. However, its effect is weaker in the lower stratosphere, where methane oxidation rates are relatively low. (**Please see P16 and L365-L372 in the revised manuscript**)

(2) Dehydration by ice PSCs, which could remove $H_2O$ from the stratosphere. However, because Arctic temperatures are generally not low enough to support the widespread formation of ice PSCs, this process plays a relatively minor role in the Arctic compared to the Antarctic. (**Please see P19 and L441-L449 in the revised manuscript**)

To further verify the relationship between $H_2O$ and the polar vortex, we present a time series of zonal-mean $H_2O$ and zonal-mean zonal wind at 30 hPa (Figure R4). SLIMCAT successfully reproduced the key characteristics of $H_2O$ variations observed by MLS, including high-$H_2O$ events in the Arctic in 2011, 2015, and 2020. In all three years, following a sharp weakening of the zonal-mean wind near 60°N (vortex breakdown), $H_2O$ concentrations dropped rapidly, suggesting a strong link between water vapor and vortex.

Unlike in the Arctic, a $H_2O$ minimum occurs every year after June in the Antarctic. This is due to the colder temperatures in Antarctica, where ice PSCs form every year, causing stratospheric dehydration. We also note that increases in zonal wind near 60°S are accompanied by decreased $H_2O$ concentrations around 45°S in the southern hemisphere winter, consistent with our conclusion that a stronger polar vortex prevents the transport of high-moisture air at high latitudes to mid-latitudes, resulting in reduced midlatitude $H_2O$. The above results of the comparison indicate that polar vortex has a significant impact on $H_2O$ near 30 hPa.

[Figure]

Figure R4. Temporal evolution of zonal-mean $H_2O$ (shading) and zonal-mean zonal wind (contours) at 30 hPa from (a) MLS observations and (b) SLIMCAT simulations.

While the paper mentions that PSC changes may affect ozone, the connection is not quantified. How much could QBO-driven PSC variability contribute to interannual ozone loss differences? The author may add a speculative estimate based on some proxy, as the volume of PSCs below a certain temperature threshold and subsequent ozone loss during spring.

**Response: Thank you for your suggestion. Previous studies have shown a strong correlation between Arctic column ozone loss and PSC volume, with a linear regression slope of 2.1 ± 0.2 DU per $10^6$ km$^3$ and a correlation coefficient of 0.96 (Rex et al., 2004). We performed vertical integration of the PSC area from CALIPSO, MIPAS, and SLIMCAT to obtain their respective PSC volumes. As shown in Table 1, the PSC volumes during the WQBO and EQBO phases and their differences are summarized. Based on the regression relationship from Rex et al. (2004), the QBO could potentially lead to an interannual variation in springtime ozone loss of approximately 8.7 DU (CALIPSO) to 46.6 DU (SLIMCAT). Here, we did not independently calculate the relationship between PSC volume and ozone loss. This is because, although PSC volume and ozone loss rate exhibit a strong linear relationship, this relationship includes not only the chemical contribution of PSCs but also the dynamical contribution. In other words, the QBO-induced interannual variation in spring ozone loss mentioned above is not solely caused by PSC changes. Therefore, a dedicated method is needed to quantify the chemical impact of PSCs on ozone depletion.**

**Table 1. PSC volumes for CALIPSO, MIPAS, and SLIMCAT during WQBO and EQBO phases and differences between the WQBO and EQBO phases (in $10^6$ km$^3$).**

|  | CALIPSO | MIPAS | SLIMCAT |
|---|---|---|---|
| WQBO | 10.82 | 19.13 | 42.29 |
| EQBO | 6.68 | 10.05 | 20.08 |
| Diff | 4.14 | 9.08 | 22.21 |

Here, we calculate the chemical ozone loss by using the "passive odd-oxygen" tracer in SLIMCAT (Feng et al., 2005). The passive tracer is set equal to $O_x = O(^3P) + O(^1D) + O_3$ (involving both chemical and dynamical processes) on the first day of the month, and it is advected passively without any chemical process. The difference between this passive odd-oxygen (OXP) and chemically integrated $O_x$ represents the chemical ozone loss ($O_3 -$ OXP, hereafter referred to as Chem $O_3$) (Wang et al., 2021).

Figure R5 and Figure R6 show the differences in $O_3$ and Chem $O_3$, respectively, between the WQBO and EQBO phases. We note that during the WQBO phase, in December and January (Figure R5a and b), there are negative $O_3$ anomalies over the Arctic on the 500–700 K isentropic levels. However, the center of this negative anomaly does not agree well with the center of the PSC area positive anomaly shown in Fig. 2 in the manuscript, suggesting that the ozone anomaly in this region is not mainly driven by chemical processes. Furthermore, in December and January (Figure R6 a and b), Chem $O_3$ exhibits positive anomalies over the Arctic on the 500–700 K isentropic levels, indicating that the observed $O_3$ decrease in this region is not due to chemical loss, but rather to dynamical processes. Starting in January, significant chemical ozone loss emerges over the Arctic, with a peak anomaly of approximately 0.05 ppmv, spatially coinciding with the PSC area positive anomaly region.

[Figure]

**Figure R5. Climatological O₃ (white contours) and the differences in zonal-mean O₃ between WQBO and EQBO phases (shadings, WQBO–EQBO) derived from SLIMCAT for the period 1979-2022. Black shading regions indicate the differences in O₃ are statistically significant at the 95% confidence level according to the Student's *t*-test.**

[Figure]

**Figure R6. Climatological Chem O₃ (white contours) and the differences in zonal-mean Chem O₃ between WQBO and EQBO phases (shadings, WQBO–EQBO) derived from SLIMCAT for the period 1979-2022. Black shading regions indicate the differences in Chem O₃ are statistically significant at the 95% confidence level according to the Student's *t*-test.**

**Figure R7 presents the vertical profiles of O₃ and Chem O₃ anomalies over the Arctic from December to March between the WQBO and EQBO phases. The chemical ozone depletion associated with PSC anomalies primarily occurs near 500 K isentropic level, which peaks in February, and is nearly zero in December. This seasonal pattern is related to solar radiation: during December, the Arctic experiences polar night, and the absence of ultraviolet radiation inhibits ozone depletion reactions. In February, the chemical ozone depletion reaches approximately 0.06 ppmv at around 480 K, while the O₃ anomaly at the same level is about 0.12 ppmv, indicating that chemical processes account for roughly 50%**

**of the total ozone loss. Although the absolute monthly chemical depletion is relatively small, its cumulative effect can result in a large impact on springtime ozone.**

[Figure]

**Figure R7. Vertical profiles of the differences in (a) $O_3$ and (b) Chem $O_3$ over the Arctic between WQBO and EQBO phases from December to March. Solid filled symbols indicate the differences are statistically significant at the 95% confidence level according to the Student's *t*-test.**

**References:**

**Rex, M., Salawitch, R. J., von der Gathen, P., Harris, N. R. P., Chipperfield, M. P., and Naujokat, B.: Arctic ozone loss and climate change, Geophys. Res. Lett., 31, L04116, https://doi.org/10.1029/2003GL018844, 2004.**

**Feng, W., Chipperfield, M. P., Roscoe, H. K., Remedios, J. J., Waterfall, A. M., Stiller, G. P., Glatthor, N., Höpfner, M., and Wang, D.-Y.: Three-dimensional model study of the Antarctic ozone hole in 2002 and comparison with 2000, Journal of the atmospheric sciences, 62, 822–837, https://doi.org/10.1175/JAS-3335.1, 2005.**

**Wang, Z., Zhang, J., Wang, T., Feng, W., Hu, Y., and Xu, X.: Analysis of the Antarctic Ozone Hole in November, Journal of Climate, 1–53,**

https://doi.org/10.1175/JCLI-D-20-0906.1, 2021.

The conclusion notes QBO disruptions under climate change but does not explore how projected QBO changes (e.g., weaker amplitude) might alter PSC trends, this may be briefly discuss this in the "Discussion and Conclusions" section.

**Response: Thank you for the insightful suggestion. In response, we have added a brief discussion to highlight how projected QBO changes may alter PSC trends. We note that a weakened QBO amplitude in the lower stratosphere may reduce its influence on Arctic temperatures, thereby reducing its effect on PSC formation. The following sentences were added in the revised paper: (Please see P24 and L548-L549 in the revised manuscript)**

*"A future weakening of the QBO amplitude (Diallo et al., 2022) may reduce its modulation of the polar vortex and temperature in the Arctic stratosphere, thereby reducing its effect on PSC variability."*

Specific comments:

- Figure 1: It could be beneficial to add to the data points a colour coding the ENSO phase, to visually show what is the possible impact on PSC area. Moreover, it would help to quantify the slopes and R2 values of the regression lines for CALIPSO and SLIMCAT.

**Response: Thank you for the helpful suggestion. In the revised Figure 1, we have added color coding to the data points based on the ENSO phase (El Niño and La Niña,), allowing for a visual assessment of ENSO's potential influence on the PSC area. Additionally, we now include the slope and $R^2$ of the regression lines for both CALIPSO and SLIMCAT, to provide a clearer quantitative comparison of the relationships. (Please see P8 and L228-L239 in the revised manuscript)**

[Figure]

*Figure 1. (a) Interannual variation of Arctic PSC volume (December–March mean)
anomalies observed by CALIPSO and MIPAS and simulated by SLIMCAT. In the
horizontal axis, blue and red labels indicate EQBO and WQBO winters, respectively.
(b, c) Arctic PSC volume (December–March mean) plotted against the QBO index in
December (left) from (b) CALIPSO and MIPAS observations and (c) SLIMCAT
simulations. Triangles represent the PSC volume simulated by SLIMCAT from 1980
to 2022, circles represent the PSC volume observed by CALIPSO from 2007 to 2021,
and squares represent the PSC volume observed by MIPAS from 2003 to 2012. Blue
markers and red markers represent the PSC volume during EQBO and WQBO,
respectively. In addition, red and blue downward-pointing triangles denote El Niño
and La Niña winters, respectively. The red lines show the linear regression of the*

*QBO index and the PSC volume for SLIMCAT and CALIPSO, respectively, with slopes (k) and coefficients of determination ($R^2$) labeled. The solid line is statistically significant at the 95% confidence level, while the dashed line is not. The probability distribution functions (PDF) of the PSC volume for the two QBO phases are shown on the right in (b) for CALIPSO and (c) for SLIMCAT.*

- Table 2: The description of "W_less $HNO_3$" and "E_more $HNO_3$" could be clearer. Specify that adding $HNO_3$ during WQBO reduces PSCs (due to less denitrification).

**Response: Thank you for your comment. We revised the descriptions in Table 2. (Please see P20 and L459-L462 in the revised manuscript)**

*Table 2. Description of the sensitivity analyses, where Δ represents the differences between the WQBO and EQBO phases.*

| Name | Change | PSC area change | Description |
|---|---|---|---|
| *W_high T* | *T-50 %×ΔT[1]* | *Decrease [2]* | *The temperature during the WQBO phase is subtracted by 50% of the temperature differences, which could raise the temperature and decrease the PSC area.* |
| *E_low T* | *T+50 %×ΔT* | *Increase* | *The temperature during the EQBO phase is added by 50% of the temperature differences, which could reduce the temperature and increase the PSC area.* |
| *W_less $H_2O$* | *$H_2O$-50 %×$ΔH_2O$* | *Decrease* | *The $H_2O$ during the WQBO phase is subtracted by 50% of the $H_2O$ differences, which could decrease the $H_2O$ concentration and the PSC area.* |
| *E_more $H_2O$* | *$H_2O$+50 %×$ΔH_2O$* | *Increase* | *The $H_2O$ during the EQBO phase is added by 50% of the $H_2O$ differences, which could increase the $H_2O$ concentration and the PSC area.* |
| *W_less $HNO_3$* | *$HNO_3$+50 %×$ΔHNO_3$* | *Decrease* | *The $HNO_3$ during the WQBO phase is added by 50% of the $HNO_3$ differences, which could decrease the $HNO_3$ concentration and the PSC area.* |

| | | | |
|---|---|---|---|
| *E_more HNO₃* | *HNO₃-50 %×ΔHNO₃* | *Increase* | *The HNO₃ during the EQBO phase is subtracted by 50% of the HNO₃ differences, which could increase the HNO₃ concentration and the PSC area.* |

- Line 20-22: It would be beneficial to clarify early that H2O anomalies have a small direct but possibly significant indirect impact via radiative cooling. See Forster and Shine (2002) for quantitative estimates.

**Response: Thank you for your valuable suggestion. We agree that stratospheric H₂O can indirectly affect PSCs through radiative cooling. However, this indirect effect is not analyzed in the current study, as it lies beyond the scope of this study. Nevertheless, we have acknowledged this limitation and discussed the potential impact of radiative cooling by H₂O on PSCs in the revised manuscript. The following sentences are added in the revised paper: (Please see P23 and L530-L538)**

*"Second, SLIMCAT does not include the chemical-radiative-dynamical coupling process. As an important trace gas in the stratosphere, H₂O not only affects chemical reactions but also contributes to the radiative cooling of the stratosphere (Bi et al., 2011). Forster and Shine (2002) showed that a 1 ppmv increase in stratospheric H₂O results in a 0.8 K decrease in the temperature of the tropical lower stratosphere, with a more pronounced cooling of 1.4 K at high latitudes. Similarly, Tian et al. (2009) found that a 2 ppmv increase in H₂O causes a temperature decrease of more than 4 K in the stratosphere at high latitudes. In particular, due to the high sensitivity of PSC formation to temperature, the indirect effects of H₂O on PSCs by influencing temperature may be comparable to its direct effects. In our sensitivity analyses, we only consider the direct effect of H₂O changes on PSCs, without accounting for the indirect impact of radiative cooling induced by H₂O anomalies. This omission may lead to an underestimation of the QBO's impact on the Arctic PSC area in Fig. 11e–h."*

- Section 2.2: A brief summary of previous validations of SLIMCAT for PSC representation would strengthen confidence. Relevant references may include Feng et al. (2021) and Li et al. (2024).

**Response: Thank you for your comment. We have revised Section 2.2 to provide a more detailed summary of the evaluation conducted by Li et al. (2024), which compared SLIMCAT results with CALIPSO observations. We also reviewed Feng et al. (2021), but found that this study does not assess the PSC representation in SLIMCAT. Therefore, we have not cited it in this context. The following sentences are revised in the revised paper: (Please see P6-P7 and L188-L189)**

*"Li et al. (2024) showed that the PSC area derived from SLIMCAT is in good agreement with CALIPSO observations in terms of seasonal evolution, interannual variability, and spatial distribution. This strengthens confidence in the performance of the SLIMCAT model in simulating PSCs."*

- Figure 4 vs. Figure 5: Please explicitly state that the ENSO exclusion does not alter the primary conclusions, but does reduce significance areas due to reduced sample size.

**Response: Thank you for your comment. The following sentences are revised in the revised paper: (Please see P13 and L306-L308)**

*"The results show similar patterns to Fig. 4, indicating that strong ENSO exclusion maintains the primary conclusions despite reducing the spatial significance extent due to reduced sample size."*

- Page 20 (Sensitivity analyses): Consider emphasizing that temperature effects dominate mainly because the Arctic stratospheric temperatures are often near PSC thresholds, making them highly sensitive (Pitts et al., 2018).

**Response: Thank you for your comment. The following sentences are rephrased in the revised paper: (Please see P21 and L468-L469)**

*"Since Arctic temperatures are concentrated around the PSC formation threshold (Fig. 7), PSCs are highly sensitive to temperature changes, and even small changes in temperature would result in significant variations in PSC."*

- Minor: Typos like "SLICMAT" instead of "SLIMCAT" (Page 4) should be corrected.

**Response: Thank you for your comment. Corrected. (Please see P4 and L108 in the revised manuscript)**